# TRUNCATED CONFORMAL PREDICTION: A SPARSITY-AWARE FRAMEWORK FOR CLASSIFICATION

## ABSTRACT

Conformal prediction (CP) is a distribution-free method for uncertainty quantification that transforms any point estimator into a set predictor. While there are many ways to enhance the efficiency of CP, an important yet underexplored way is to incorporate domain prior knowledge into the algorithm. In this paper, we focus on leveraging a sparsity prior into CP algorithm for classification tasks. Specifically, the probability simplex often exhibits a sparsity structure in large-scale classification tasks. However, existing classifiers typically include a softmax layer that diminishes this sparsity prior. To address this issue, we propose a truncation-normalization operator that uses a sparsity prior in CP, thereby improving efficiency. Both theoretical and empirical results reveal the following insights: (i) the U-shaped relation between set size and truncation level ensures the existence of a nonzero optimal truncation level; (ii) the oracle set could be recovered by choosing the optimal truncation level, which is unattainable without truncation; and (iii) optimal truncation level correlates positively with model quality.

## 1 INTRODUCTION

As a methodology aiming at identifying and characterizing uncertainty in machine learning models, uncertainty quantification plays a vital role in high-stake decision-making scenarios, such as drug development in medicine (Laghuvarapu et al., 2023; Nolte et al., 2024), risk analysis in finance (Bogani et al., 2024; Kato, 2024), and may even enhance the trustworthiness of AI systems by estimating the confidence of predictions and addressing issues like hallucinations in large language models (Yadkori et al., 2024; Li et al., 2023).

Among various methods for uncertainty quantification, conformal prediction (CP) has gained popularity due to its ease of implementation and finite-sample coverage guarantees (Vovk et al., 2005; Shafer & Vovk, 2008; Lei et al., 2015). Its core principle is to transform any machine learning model's point estimator into a reliable set predictor. Notably, the efficiency of a CP algorithm is typically measured by the size of the resulting prediction set, which is related to the quality of the first-stage point estimator. Consequently, numerous methods have been proposed to improve efficiency by enhancing the accuracy of the point estimator (Noorani et al., 2024; Stutz et al., 2021b).

In this paper, we introduce an alternative yet underexplored way for improving efficiency in classification tasks: *incorporating domain prior knowledge into the CP algorithm*. An important prior for the classification task is the *sparsity structure* (Definition 1) in the probability vector, which is prevalent in practice. For example, if a sample is a cat, which belongs to the broader class of animals, it hardly belongs to unrelated classes such as fruits or tools. Consequently, the probabilities assigned to these unrelated classes should be zero. But in practice, to ensure that the estimated probability $\hat{\pi}(\boldsymbol{x})$ produces values within the probability simplex, a softmax operator is typically applied. This operation tends to diminish the sparsity prior inherent in the true probability vector $p(y \mid \boldsymbol{x})$.

Unfortunately, the efficiency of CP is sensitive to the entire distribution of the estimated probabilities. Therefore, a sparsity-aware estimator that recovers the sparsity structure in the distribution potentially improves efficiency. In comparison, without incorporating this sparsity structure, small noise[1] values within the estimated probability vector may inflate the prediction set size, as meeting the same threshold may require including more classes with low probabilities, illustrated in fig. 1. Thus, this paper seeks to address the following question:

*When and how can the efficiency of CP algorithm be improved by incorporating a sparsity prior?*

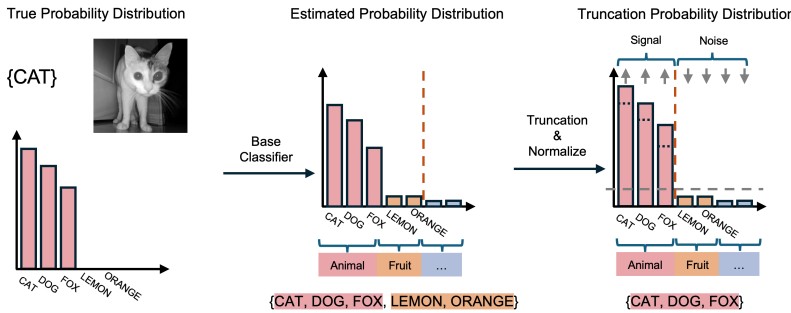

Figure 1: Framework of truncated CP (Algorithm 1). To mitigate the impact of small noise in the estimated probability vector, predicted probabilities below a threshold are truncated to zero, and the remaining values are renormalized to ensure they lie within the probability simplex.

To address this question, we propose a truncation framework (fig. 1) to incorporate the sparsity prior structure into the CP algorithm. This approach operates on the estimated probability vector $\hat{\pi}(\cdot)$, by setting values below a threshold to zero and then normalizing to ensure the result lies within the probability simplex. This framework offers two key advantages: First, it is simple yet effective and preserves the post-hoc nature of CP. Second, it imposes no restrictions on the choice of score functions, making it broadly compatible with existing classification CP algorithms. Despite these advantages, several questions remain, both in practice and theory: **(i)** how to determine a valid truncation level, **(ii)** whether the optimal prediction set size could be achieved, and **(iii)** how the optimal truncation level relates to first-stage model quality.

To address these considerations, we conduct both theoretical analyses (Section 4) and empirical experiments (Section 5), yielding the following insights: **Firstly**, we establish a U-shaped relation between the prediction set size and truncation level (Theorem 4). Specifically, as the truncation level increases, the size of the prediction set initially decreases and then increases. This finding provides practical guidance for selecting the truncation level at the "elbow" position (Corollary 5), which effectively separates signal from noise[1] in the estimated probability vector, balancing compact prediction sets with group coverage. **Secondly**, by incorporating the sparsity prior, our proposed truncation method asymptotically recovers the oracle set predictor under a consistent first-stage classifier with APS score function (Theorem 7). As a comparison, we further demonstrate that this asymptotic property is unattainable without truncation, providing two examples in Section 4.4. **Finally**, the results in fig. 3 further reveal how first-stage model quality relates to the efficiency improvements from truncation. Specifically, the "elbow" position shifts upward with training epochs as improved model quality widens the gap between signal and noise, enabling better truncation quality and an effective prediction set. Our contributions are further summarized in Appendix A.1.

## 2 RELATED WORK

Conformal prediction is a statistical framework for uncertainty quantification, ensuring finite-sample coverage guarantees with a post-hoc approach (Vovk et al., 2005; Shafer & Vovk, 2008; Lei et al., 2018; Angelopoulos & Bates, 2021). Its flexibility has facilitated applications in diverse fields, including classification (Romano et al., 2020b; Angelopoulos et al., 2020), regression (Lei et al., 2018; Romano et al., 2019), graph neural networks (Huang et al., 2023b; Zargarbashi et al., 2023), large language models (Quach et al., 2023; Yadkori et al., 2024), time series analysis (Xu & Xie, 2021; Zaffran et al., 2022), and survival analysis (Candès et al., 2023; Teng et al., 2021).

A central research direction of CP in classification tasks is the development of effective score functions to enhance efficiency and adaptiveness. For example, LAC (Sadinle et al., 2019) achieves small prediction set sizes but offers limited conditional coverage. To address this, APS (Romano et al., 2020a) introduces input-adaptive score adjustments to improve conditional coverage, though at the expense of efficiency. RAPS (Angelopoulos et al., 2020) further advances this by penalizing hard examples, thereby reducing their impact on the score and striking a balance between efficiency and

---

[1]In this paper, signal refers to the minimum estimated probability in $\hat{\pi}(\boldsymbol{x})$ among the non-zero entries of $p(y \mid \boldsymbol{x})$, noise refers to entries with true probability zero that are estimated as non-zero in $\hat{\pi}(\boldsymbol{x})$.

conditional coverage. Additionally, rank-based score functions such as SAPS (Huang et al., 2023a) and RANK (Luo & Zhou, 2024) have emerged as notable contributions in this domain.

Beyond the design of score functions, another avenue for improving efficiency involves modifying the model obtained in the first stage. This is achieved through two main strategies: altering the training process or employing fine-tuning methods. The first strategy optimizes the first-stage model using uncertainty-aware conformal loss functions (Stutz et al., 2021a; Einbinder et al., 2022; Correia et al., 2024), although this compromises the post-hoc nature of CP. To address this limitation, the second strategy introduces fine-tuning methods that leverage a calibration process on the first-stage model's output, while keeping the original model unchanged (van der Laan & Alaa, 2024).

Incorporating prior information into machine learning algorithms may improve model generalization and align outcomes with domain-specific knowledge in various scenarios. For example, known spurious correlations between labels are exploited to steer the learner toward truly invariant features (Arjovsky et al., 2019). Equivariance is enforced on rotations, translations, reflections, and permutations within graph neural networks to boost molecular-property prediction (Satorras et al., 2021). Classifier gradients are injected into the reverse diffusion process to condition image generation on class information (Dhariwal & Nichol, 2021). However, incorporating prior information into machine learning methods remains underexplored in the context of CP algorithms.

## 3 SPARSITY PRIOR AND CONFORMAL PREDICTION IN CLASSIFICATION

In this section, we introduce two prevalent sparsity structures and several baseline conformal prediction algorithms commonly employed in classification tasks.

**Notations.** Let $\boldsymbol{x}[i]$ denote the $i$-th element in the vector $\boldsymbol{x}$, and $\mathbb{I}_{\{\cdot\}}$ represent the indicator function. Consider a classification task with $K$ classes, where each data point $(\boldsymbol{x}, y) \in \mathcal{X} \times \mathcal{Y}$, with $\mathcal{X} = \mathbb{R}^p$ and $\mathcal{Y} = \{1, \ldots, K\}$. For any $\boldsymbol{x} \in \mathcal{X}$, the conditional distribution of $(\boldsymbol{x}, y)$ is denoted by $\pi_y(\boldsymbol{x}) = \mathbb{P}[Y = y \mid \boldsymbol{X} = \boldsymbol{x}]$ for each $y \in \mathcal{Y}$, and the vector $\pi(\boldsymbol{x}) = [\pi_1(\boldsymbol{x}), \ldots, \pi_K(\boldsymbol{x})] \in \Delta^{K-1}$, where $\Delta^{K-1} := \{\boldsymbol{\pi} \in \mathbb{R}^K : \boldsymbol{\pi}[i] \geq 0, i = 1, \ldots K; \boldsymbol{\pi}[1] + \ldots + \boldsymbol{\pi}[K] = 1\}$. The order statistics of $\{\pi_1(\boldsymbol{x}), \ldots, \pi_K(\boldsymbol{x})\}$ are denoted as $\pi_{(1)}(\boldsymbol{x}) \geq \pi_{(2)}(\boldsymbol{x}) \geq \ldots \geq \pi_{(K)}(\boldsymbol{x})$. There are two prevalent structures in classification tasks: *sparsity* and *group sparsity*.

**Definition 1** (Sparsity Structure). *A conditional probability vector $\pi(\boldsymbol{x}) \in \Delta^{K-1}$ exhibits sparsity if at least one of its entries is zero, and the sparsity ratio is defined as $\frac{1}{K} \sum_{k=1}^{K} \mathbb{I}_{\{\pi_k(\boldsymbol{x})=0\}}$.*

Sparsity serves as prior information by reducing the candidate label set. It restricts the space in which $\pi(\boldsymbol{x})$ lies, thereby simplifying the classification task. This property is prevalent in high-dimensional problems, where $K$ is large but only a small subset is relevant for any given sample.

**Definition 2** (Group Sparsity Structure). *A conditional distribution $\mathbb{P}(Y|\boldsymbol{X})$ is group sparse if there exists a partition $\{\mathcal{L}_1, \ldots, \mathcal{L}_G\}$ of the label set $\mathcal{Y}$ such that, for any $\boldsymbol{x}$ in group $g \in \{1, \ldots, G\}$, we have $\mathrm{supp}(\mathbb{P}(Y \mid \boldsymbol{X} = \boldsymbol{x})) \subseteq \mathcal{L}_g$, where $\mathrm{supp}(\cdot)$ denotes the support set of a distribution.*

Group sparsity is a stronger assumption than sparsity. Intuitively, it introduces a hierarchical structure into the classification task. For example, suppose $K = 6$ labels represent different categories: $\{1 = \mathrm{dog}, 2 = \mathrm{cat}, 3 = \mathrm{horse}, 4 = \mathrm{car}, 5 = \mathrm{truck}, 6 = \mathrm{bus}\}$. We can partition them into two groups: $\mathcal{L}_1 = \{1, 2, 3\}$ (animals), $\mathcal{L}_2 = \{4, 5, 6\}$ (vehicles).

Let $\hat{\pi}(\boldsymbol{x}) = \sigma(\boldsymbol{z}(\boldsymbol{x}))$ represent the estimated probability simplex produced by a base model for a feature $\boldsymbol{x}$, where $\boldsymbol{z}(\boldsymbol{x}) \in \mathbb{R}^K$ is the logit vector (*i.e.*, the pre-softmax output) and $\sigma(\cdot)$ denotes the softmax operator, defined as $\sigma_i(\boldsymbol{z}) = \exp(\boldsymbol{z}[i]) / \sum_{j=1}^{K} \exp(\boldsymbol{z}[j]), i = 1, \ldots, K$. For simplicity, we assume that the non-zero entries of the probability simplices $\hat{\pi}(\cdot)$ and $\pi(\cdot)$ contain no ties. If ties occur in the label ordering, they could be resolved by introducing randomness.

**Conformal Prediction in Classification Tasks.** Conformal prediction is a distribution-free technique that constructs a prediction set of classes $\mathcal{C}_\alpha(\boldsymbol{x})$ for a given sample $\boldsymbol{x} \in \mathcal{X}$ in a classification task, ensuring the true class $y \in \mathcal{C}_\alpha(\boldsymbol{x})$ with a probability of $1 - \alpha$, where $\alpha \in (0, 1)$ is a predefined significance level (Vovk et al., 1999; Papadopoulos et al., 2002). To achieve this statistical guarantee, CP exploits the exchangeability of a holdout calibration set $\{(\boldsymbol{x}_i, y_i)\}_{i=1}^{n}$ to create a decision rule based on a user-specified score function. Specifically, CP introduces a

---

**Algorithm 1** Truncation Framework for Classification CP

---

1: **Input:** Score function $s(\cdot, \cdot; \cdot)$, estimated probability $\hat{\pi}(\cdot)$, truncation level $\lambda$, calibration set $\{(\boldsymbol{x}_i, y_i)\}_{i=1}^n$, confidence level $\alpha$, a new input $\boldsymbol{x}_{n+1}$.
2: Sparsify the estimated conditional probability simplex $\hat{\pi}(\cdot)$ to obtain $\hat{\pi}^\lambda(\cdot)$ by applying the transformation in equation 2 with the truncation level $\lambda$;
3: Compute the score on calibration set $\{s(\boldsymbol{x}_i, y_i; \hat{\pi}^\lambda)\}_{i=1}^n$ based on $\hat{\pi}^\lambda(\cdot)$;
4: Compute the $\lceil (1-\alpha)(1+n) \rceil$-largest value in $\{s(\boldsymbol{x}_i, y_i; \hat{\pi}^\lambda)\}_{i=1}^n$ as $\hat{q}^\lambda$;
5: **Output:** The prediction set $\mathcal{C}(\boldsymbol{x}_{n+1}; \hat{\pi}^\lambda, \hat{q}^\lambda) = \{y : s(\boldsymbol{x}_{n+1}, y; \hat{\pi}^\lambda) \le \hat{q}^\lambda\}$.

---

heuristic score function $s(\boldsymbol{x}, y; f) \in \mathbb{R}$ based on $f : \mathcal{X} \to \Delta^{K-1}$, where a higher score indicates a lower level of agreement between $\boldsymbol{x}$ and $y$, and the $\frac{\lceil (1-\alpha)(1+n) \rceil}{n}$-quantile $\hat{q}$ of the scores $\{s(\boldsymbol{x}_1, y_1; f), \dots, s(\boldsymbol{x}_n, y_n; f)\}$ is subsequently computed. Then for a new input $\boldsymbol{x}_{n+1}$, the prediction set is defined as $\mathcal{C}(\boldsymbol{x} = \boldsymbol{x}_{n+1}; f, \tau = \hat{q}) = \{y : s(\boldsymbol{x}_{n+1}, y; f) \le \hat{q}\}$. [2]

The design of the score function is crucial in CP, with many prominent designs proposed to enhance the adaptiveness and efficiency of CP. In this paper, we focus on the following methods, with a more detailed discussion provided in Appendix A.2. In APS, (Romano et al., 2020a), score function

$$s(\boldsymbol{x}, y; \hat{\pi}) = \sum_{y'=1}^{K} \hat{\pi}_{y'}(\boldsymbol{x}) \mathbb{I}_{\{\hat{\pi}_{y'}(\boldsymbol{x}) \ge \hat{\pi}_y(\boldsymbol{x})\}} \tag{1}$$

is designed to improve the conditional coverage. To improve efficiency, RAPS (Angelopoulos et al., 2020) defines the score as $s(\boldsymbol{x}, y; \hat{\pi}) = \sum_{y'=1}^{K} \hat{\pi}_{y'}(\boldsymbol{x}) \mathbb{I}_{\{\hat{\pi}_{y'}(\boldsymbol{x}) \ge \hat{\pi}_y(\boldsymbol{x})\}} + \gamma(o_{\boldsymbol{x}}(y) - k_{\text{reg}})_+$, where $o_{\boldsymbol{x}}(y)$ is the index to which label $y$ is mapped after the scores are sorted, and $(\cdot)_+ := \max\{\cdot, 0\}$.

# 4 TRUNCATED CONFORMAL PREDICTION AND THEORETICAL GUARANTEES

In this section, we introduce a truncation framework to incorporate a sparsity prior into CP algorithm for classification. We first present the truncated CP algorithm in Section 4.1, which removes small noise from the estimated probability vector for any given truncation level. As the truncation level directly governs the balance between signal and noise, a theoretical analysis based on the APS score function in Section 4.2 is provided to explain why the prediction set size initially decreases and then increases as the truncation level grows. Subsequently, we demonstrate that truncated CP achieves valid finite sample coverage and asymptotically recovers the oracle set predictor under a consistent first-stage classifier (Section 4.3), which is unattainable without truncation. Finally, two examples in Section 4.4 further illustrate the differences between using and not using truncation.

## 4.1 TRUNCATION FRAMEWORK

In this section, we propose the truncated CP algorithm to incorporate the sparsity prior into the uncertainty set. In large-scale classification tasks, the probability simplex $p(y|\boldsymbol{x})$ often exhibits sparsity structures, and it is challenging to incorporate this prior information via score function design (see Section 4.4). A more effective alternative is to address sparsity in the first stage by constructing the estimator $\hat{\pi}(\cdot)$ with an explicit sparsity guarantee. To this end, we propose to recover the sparsity prior in $\hat{\pi}(\cdot)$ by the following truncation-normalization operator.

**Truncation-Normalization Operator.** We truncate predicted probabilities below a threshold to zero and subsequently normalize the result to ensure that they lie within the probability simplex. Specifically, for a given truncation level $\lambda$, the sparsified estimated probability simplex is defined via a truncation-normalization operator $T_\lambda(\cdot)$ as:

$$\hat{\pi}_y^\lambda(\boldsymbol{x}) = T_\lambda(\hat{\pi}(\boldsymbol{x}))_y := \begin{cases} \hat{\pi}_y(\boldsymbol{x})/\rho^\lambda(\boldsymbol{x}), & \text{if } \hat{\pi}_y(\boldsymbol{x}) > \min\left(\lambda, \hat{\pi}_{(2)}(\boldsymbol{x})\right), \\ 0, & \text{otherwise,} \end{cases} \tag{2}$$

---

[2]To ensure valid coverage in classification tasks, a random variable $U \sim \text{Unif}(0, 1)$ is often incorporated into $\mathcal{C}$. Without loss of generality, we omit it in this paper for simplicity.

where $\rho^\lambda(\boldsymbol{x}) = \sum_{y'=1}^K \hat{\pi}_{y'}(\boldsymbol{x}) \, \mathbb{I}_{\left\{ \hat{\pi}_{y'}(\boldsymbol{x}) > \min\left(\lambda, \hat{\pi}_{(2)}(\boldsymbol{x})\right) \right\}}$ is a normalizing factor that ensures that the result remains a probability simplex. The proposed truncation-normalization operator offers two benefits: *(i) it reduces the impact from small noises and (ii) amplifies the variance among signals with a normalizing factor smaller than 1*. Both of these potentially improve efficiency. This operator forms the core of our Truncated CP algorithm, which is introduced below.

**Truncated Conformal Prediction Algorithm.** Once the sparsity prior is incorporated into the base classifier model $\hat{\pi}(\cdot)$ through the truncation-normalization operator $T_\lambda(\cdot)$, the prediction set is established using the CP procedure based on a user-specified score function, and the entire process is summarized in Algorithm 1. Notably, the truncation framework preserves the post-hoc nature of CP since the operator $T_\lambda(\cdot)$ only applies to the output of the base classifier $\hat{\pi}(\cdot)$.

**Choice of the Truncation Level.** One key challenge in our truncation framework lies in the selection of the truncation level $\lambda$. To address this, we empirically examine how $\lambda$ influences both the size of the prediction set and the group coverage using a holdout validation set. As illustrated in fig. 2, the size of the prediction set decreases initially with an increasing $\lambda$, but starts to grow after reaching a certain point. This empirical observation provides practical guidance: $\lambda$ *is recommended to be selected near the "elbow" point using a holdout set in calibration data*, which achieves a balance between compact prediction sets and preserving conditional coverage (see Appendix C.1 for details). In Section 4.2, we provide a theoretical explanation for this phenomenon (Corollary 5).

**Remark 3.** From another perspective, truncated CP implicitly modifies the score function from $s(\boldsymbol{x}, y; \hat{\pi})$ to $s(\boldsymbol{x}, y; \hat{\pi}^\lambda)$. It incorporates the sparsity prior into the CP algorithm by adjusting the estimated probability vector $\hat{\pi}(\cdot)$, without explicitly designing a new score function.

## 4.2 IMPACT OF TRUNCATION LEVEL ON SIZE

In this section, we provide a theoretical explanation (Theorem 4) for the phenomenon empirically observed in fig. 2: the size of the prediction set initially decreases as the truncation level $\lambda$ increases, but begins to grow once $\lambda$ surpasses a certain threshold. This behavior could be explained by the relationship between the signal and noise in the estimated probability vector $\hat{\pi}(y \mid \boldsymbol{x})$. Moreover, when the truncation level is chosen to balance these components for a given test point, the size of the prediction set could be effectively reduced, with practical guidance provided in Corollary 5.

**Theorem 4** (Impact of Truncation Level on Size). *For any* $\boldsymbol{x} \in \mathcal{X}$, *let* $\hat{\pi}(\boldsymbol{x}) = \sigma(\boldsymbol{z}(\boldsymbol{x})) \in \Delta^{K-1}$ *represent the estimated probability, where* $\boldsymbol{z}$ *is the logit vector output by a base model. For a truncation level* $\lambda > 0$, *the sparsified probability vector for* $\boldsymbol{x}$ *is defined as* $\hat{\pi}^\lambda(\boldsymbol{x}) = T_\lambda(\hat{\pi}(\boldsymbol{x}))$, *as given in eq.* (2). *Let* $\hat{q}^\lambda$ *denote the quantile of the score computed on the calibration set. The size of the prediction set for* $\boldsymbol{x}$ *is given by* $L_\lambda = L(\boldsymbol{x}; \hat{\pi}^\lambda, \hat{q}^\lambda) = \min \left\{ c \in \{1, \ldots, K\} : \sum_{i=1}^c \hat{\pi}_{(i)}^\lambda(\boldsymbol{x}) \geq \hat{q}^\lambda \right\}$. *The number of non-zero elements in* $\hat{\pi}^\lambda(\boldsymbol{x})$ *is* $S_\lambda(\boldsymbol{x}) = \max_i \{ i : \hat{\pi}_y(\boldsymbol{x}) > \min(\lambda, \hat{\pi}_{(2)}(\boldsymbol{x})) \}$. *The gap function is defined as* $g(\boldsymbol{z}; \lambda, M) = \sum_{i=1}^M [\sigma_i(\boldsymbol{z}) - T_\lambda(\sigma(\boldsymbol{z})_i)]$. *Let* $\boldsymbol{z}[-1] = (\boldsymbol{z}[2], \ldots, \boldsymbol{z}[K])$ *and* $\Delta \boldsymbol{z} = \boldsymbol{z}[1] - \boldsymbol{z}[2]$. *Under the following assumptions:*
*(C1) There exists* $\delta$ *such that* $\hat{\pi}_y(\boldsymbol{x}) > \delta$ *for any sample* $(\boldsymbol{x}, y)$.
*(C2)* $\hat{q}^0$ *and* $\hat{q}^\lambda$ *are derived from the same sample* $\boldsymbol{x}^q$ *in the calibration set.*
*(C3)* $\sum_{i=S_\lambda(\boldsymbol{x}^q)+1}^K \hat{\pi}_{(i)}(\boldsymbol{x}^q) = o(\gamma)$.
*(C4)* $\boldsymbol{z}^q[1] > \boldsymbol{z}[1]$, *where* $\boldsymbol{z}^q$ *is the logit vector of* $\boldsymbol{x}^q$. *For any* $\boldsymbol{z}' = \theta \boldsymbol{z}^q[1] + (1 - \theta) \boldsymbol{z}[1]$ $(\theta \in [0, 1])$ *and* $M \leq L_\lambda$, $|\nabla g_{\boldsymbol{z}[1]}(\boldsymbol{z}'; \lambda, M)(\boldsymbol{z}^q[1] - \boldsymbol{z}[1])| > |\nabla g_{\boldsymbol{z}[-1]}(\boldsymbol{z}'; \lambda; \lambda, M)^\top (\boldsymbol{z}^q[-1] - \boldsymbol{z}[-1])|$.
*(C5) The quantity* $\Delta \boldsymbol{z}$ *increases monotonically with the score* $s(\boldsymbol{x}, y; \hat{\pi})$.
*Then, for any truncation level* $\lambda \leq \delta$, *it holds that:*

$$\begin{cases} \text{If} \quad \lambda < \frac{1}{K}, \exp(\Delta \boldsymbol{z}) > (K-1)(1 - K\lambda) \Longrightarrow L_0 \geq L_\lambda, \\ \text{If} \quad \lambda \geq \frac{1}{K} \Longrightarrow L_0 \geq L_\lambda. \end{cases}$$

Theorem 4 provides insight into when truncation may reduce the size. Specifically, the truncation level should not exceed the signal in $\hat{\pi}(\boldsymbol{x})$. This implicitly requires a separation between the signal and noise in $\hat{\pi}(\boldsymbol{x})$. Such a separation becomes more evident as the quality of the first-stage estimator improves (see fig. 3). Consequently, as the truncation level increases, it initially restores sparsity and reduces the size. However, once the truncation level reaches the signal value, it starts to disrupt the signal, leading to an increasing size. For example, when $\lambda = 1$, the truncation nullifies all but the

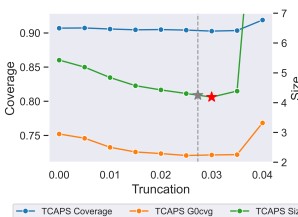 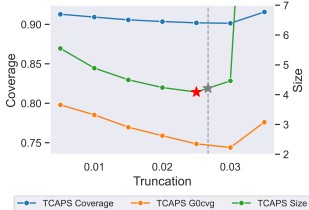 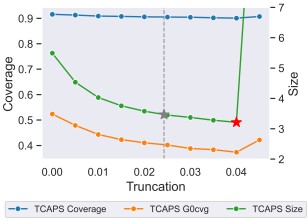

(a) Group Size: [10,20,30] 1-1-1[3]  (b) Group Size: [10,20,30] 2-2-2  (c) Group Size: [10,20,30] 3-3-3

Figure 2: Line plots of group coverage and size versus truncation level, with coverage fixed at $1-\alpha$. **As the truncation level increases, the set size under the coverage constraint initially decreases and then increases, forming an "elbow" region.** The optimal truncation level (red star) can be selected via a holdout validation set (Corollary 5), while a sub-optimal level (horizontal dashed line) can be chosen without validation (using $\psi = 2/5\alpha$ in Remark 6), as detailed in Appendix C.2.

top component in $\hat{\pi}(\cdot)$, causing the prediction set to degenerate into the full label set $\mathcal{Y}$ to maintain coverage. This analysis provides practical guidance for selecting a valid $\lambda$ in practice.

**Corollary 5** (Optimal Truncation Level). *Under conditions (C1-C5) in Theorem 4, the prediction set size first decreases and then increases as the truncation level increases. Thus, the optimal truncation level $\lambda^*$ could be selected using a holdout validation set to minimize the prediction set size.*

**Remark 6** (Sub-optimal Truncation Level). To determine the optimal truncation level, a holdout validation set is typically required. A more data-efficient but sub-optimal alternative is to exploit the fact that the optimal truncation level lies between the signal and noise, as demonstrated in Theorem 4. Specifically, for any given significance level $\alpha$, a sub-optimal truncation level could be chosen as the $\lceil (1-\alpha+\psi)(1+n) \rceil$-largest value in $\{\hat{\pi}_{y_i}(\boldsymbol{x}_i)\}_{i=1}^n$ using the calibration set, where $\psi < \alpha$ controls the implicit assumption that $(\alpha - \psi)\%$ of the samples predicted by the base model belong to the noise set (see Appendix C.2 for details). In practice, the default value of $\psi$ is recommended to lie within the range $[2/5\alpha, 4/5\alpha]$, with larger values preferred for higher-quality models.

**Explanations of Conditions.** (C1) specifies that the truncation level should not exceed the signal, ensuring that only noise is truncated. (C2) assumes that both $\hat{q}^0$ and $\hat{q}^\lambda$ correspond to the same sample in the calibration set, which is reasonable as the truncation operation preserves the rank of label probabilities, causing minimal changes to the resulting score ranks within the calibration set, empirically demonstrated in fig. 4. (C3) ensures that the cumulative noise being truncated is small compared to the signal. This condition is achievable when the model includes a softmax layer, which amplifies the gap between signal and noise when the model is well-trained. (C4) and (C5) are technical conditions that are similarly employed in Dabah & Tirer (2024). These conditions assume that the gap function $g$ is dominated by the difference between the largest logit, with such dominance becoming more pronounced as the score increases, validated in fig. 5.

## 4.3 SIZE AND COVERAGE GUARANTEE

In this section, we establish a theoretical guarantee (Theorem 7) that the oracle set predictor could be asymptotically recovered after truncation, assuming the first-stage estimator is consistent. Moreover, the finite sample coverage remains valid for the truncated CP algorithm (Theorem 8).

**Theorem 7** (Asymptotic Oracle Recovery). *For any $\boldsymbol{x} \in \mathcal{X}$, let $\pi(\boldsymbol{x})$ and $\hat{\pi}(\boldsymbol{x})$ denote the true and estimated probability simplex, where $\pi(\boldsymbol{x})$ has sparsity structure. Let the score function be the APS score defined in eq. (1). Denote the prediction set obtained by $\hat{\pi}(\cdot)$ after truncation with the level $\lambda$ as $\mathcal{C}_{\mathrm{TCAPS}}(\boldsymbol{x}; \hat{\pi}^\lambda, \hat{q}^\lambda) = \{y : s(\boldsymbol{x}, y; \hat{\pi}^\lambda) \le \hat{q}^\lambda\}$, and the oracle prediction set based on $\pi(\cdot)$ as $\mathcal{C}^{\mathrm{oracle}}(\boldsymbol{x}; \pi, q) = \{y : s(\boldsymbol{x}, y; \pi) \le q\}$, where $\hat{q}^\lambda$ and $q$ are the $\lceil (1-\alpha)(1+n) \rceil$-largest value of scores computed on the calibration set $\{(\boldsymbol{x}_i, y_i)\}_{i=1}^n$ using $\hat{\pi}^\lambda$ and $\pi$. Let $S_{\boldsymbol{x}} = \{i : \pi_i(\boldsymbol{x}) > 0\}$ be the support of $\pi(\boldsymbol{x})$, and $S_{\boldsymbol{x}}^C = \{i : \pi_i(\boldsymbol{x}) = 0\}$. Assume there exists $\delta$ such that $\min_{i \in S_{\boldsymbol{x}}} \hat{\pi}_i(\boldsymbol{x}) > \delta > \max_{i \in S_{\boldsymbol{x}}^c} \hat{\pi}_i(\boldsymbol{x})$ for all $\boldsymbol{x}$, and $\|\hat{\pi}(\boldsymbol{x}) - \pi(\boldsymbol{x})\|_2 \to 0$ as $n \to \infty$. Choosing $\lambda = \delta$ yields*

$$\mathcal{C}_{\mathrm{TCAPS}}(\boldsymbol{x}; \hat{\pi}^\lambda, \hat{q}^\lambda) \to \mathcal{C}^{\mathrm{oracle}}(\boldsymbol{x}; \pi, q) \text{ as } n \to \infty.$$

*In contrast, the prediction set $\mathcal{C}(\boldsymbol{x}; \hat{\pi}, \hat{q})$ constructed without truncation is larger than the oracle set, i.e., $|\mathcal{C}(\boldsymbol{x}; \hat{\pi}, \hat{q})| > |\mathcal{C}^{\mathrm{oracle}}(\boldsymbol{x}; \pi, q)|$ for any sample size $n$.*

Theorem 7 addresses the efficiency limitations of conformal prediction when the sparsity prior is not incorporated, even with a consistent first-stage estimator (see examples in Section 4.4). Specifically, when the truncation level is properly chosen to separate signal from noise, the sparsity prior is restored by the truncation method, eliminating the impact of small noise on $S_{\boldsymbol{x}}^c$ that affects the size, thereby asymptotically recovering the oracle set predictor. Besides the asymptotic oracle recovery property, the finite-sample coverage property remains valid after truncation, as shown in Theorem 8.

**Theorem 8** (Coverage Guarantee for Truncated CP). *Suppose $\{(\boldsymbol{x}_i, y_i)\}_{i=1}^n$ and $(\boldsymbol{x}_{n+1}, y_{n+1})$ are i.i.d. and let $\mathcal{C}(\boldsymbol{x}_{n+1}; \hat{\pi}^\lambda, \hat{q}^\lambda)$ be the prediction set of $\boldsymbol{x}_{n+1}$ obtained from Algorithm 1 by given confidence level $\alpha$. The following coverage guarantee holds: $\mathbb{P}\left(y_{n+1} \in \mathcal{C}\left(\boldsymbol{x}_{n+1}; \hat{\pi}^\lambda, \hat{q}^\lambda\right)\right) \geq 1 - \alpha$.*

### 4.4 COMPARISON OF EXAMPLES WITH AND WITHOUT TRUNCATION

In this section, we present two examples to compare the performance of CP with and without truncation. Compared to approaches that solely modify the score function without truncation, our framework improves the efficiency of CP. Specifically, **(i)** under the sparsity structure (Definition 1), small values in the estimated probability vector inflate the prediction set size under APS, even when the classifier achieves perfect accuracy. **(ii)** Under the stronger group sparsity structure (Definition 2), when the sparsity ratio depends on the input, RAPS fails to capture this relationship, resulting in reduced efficiency. We begin with a general conjecture on efficiency improvement.

**Conjecture 9** (Efficiency is Improved after Truncation under Sparsity Prior). The truncation operation improves the efficiency of CP under both sparsity and group sparsity structures.

To provide a clearer intuition of Conjecture 9, we focus on two simple yet representative special cases in Example 10, 11, leaving the general but more involved derivation for future work.

**Example 10** (APS is Inefficient Under Sparsity Structure). *Let $\boldsymbol{x} \in \mathbb{R}^2$, $\mathcal{Y} = \{1, 2, 3, 4\}$, and $\{(\boldsymbol{x}_i, y_i)\}_{i=1}^n$ be a calibration set, with the underlying true probability defined as: $\pi_{(1)}(\boldsymbol{x}) = 1$, $\pi_{(2)}(\boldsymbol{x}) = \pi_{(3)}(\boldsymbol{x}) = \pi_{(4)}(\boldsymbol{x}) = 0$. Suppose the estimated probability of the input $\boldsymbol{x}$ provided by the model is given as follows: $\hat{\pi}_{(1)}(\boldsymbol{x}) = 1 - 3\xi \cdot \varepsilon$, $\hat{\pi}_{(2)}(\boldsymbol{x}) = \hat{\pi}_{(3)}(\boldsymbol{x}) = \hat{\pi}_{(4)}(\boldsymbol{x}) = \xi \cdot \varepsilon$, where $\varepsilon \sim \text{Bernoulli}(p)$ and $\xi < 0.25$.*

*Then the average prediction set size produced by truncated CP is 1, which is smaller than $1 + 3p \cdot \mathbb{P}(B < \lceil(1-\alpha)(1+n)\rceil)$ without truncation, where $B = \sum_{i=1}^n \epsilon_i \sim \text{Binomial}(n, p)$.*

To simulate sparsity, we assign zero values to 75% of the indices in $\pi(\cdot)$ as zero, and set $K = 4$ for simplicity. Notably, the point estimator based on $\hat{\pi}(\cdot)$ has 100% classification accuracy. Without truncation, the APS score is computed as $s(\boldsymbol{x}, y; \hat{\pi}) = \sum_{y'=1}^K \hat{\pi}_{y'}(\boldsymbol{x})\mathbb{I}_{\{\hat{\pi}_{y'}(\boldsymbol{x}) \geq \hat{\pi}_y(\boldsymbol{x})\}} = \hat{\pi}_{(1)}(\boldsymbol{x}) = 1 - 3\xi \cdot \varepsilon$. Then, the quantile of score computed on the calibration set $\{s(\boldsymbol{x}_i, y_i; \hat{\pi})\}_{i=1}^n$ is given by:

$$\hat{q} = \begin{cases} 1 - 3\xi, & \text{if } B \geq \lceil(1-\alpha)(1+n)\rceil, \\ 1, & \text{if } B < \lceil(1-\alpha)(1+n)\rceil. \end{cases} \tag{3}$$

When the quantile $\hat{q} = 1$, the prediction set of new input $\boldsymbol{x}$ by APS is $\{1, 2, 3, 4\}$ with probability $p$. This result highlights that APS may produce a large prediction set when sparsity is not accounted for. More details are presented in the Appendix B.4. This issue could be addressed by incorporating the sparsity prior into $\hat{\pi}(\cdot)$ through truncation, *i.e.*, imposing these small noise terms $\xi\epsilon$ into 0.

**Example 11** (RAPS is Inefficient Under Group Sparsity Structure). *Let $\boldsymbol{x} \in \mathbb{R}^2$, $\mathcal{Y} = \{1, 2, 3, 4\}$, and $\{(\boldsymbol{x}_i, y_i)\}_{i=1}^n$ be a calibration set, with the underlying true probability defined as:*

$$\begin{cases} \pi_{(1)}(\boldsymbol{x}) = 1, & \pi_{(2)}(\boldsymbol{x}) = \pi_{(3)}(\boldsymbol{x}) = \pi_{(4)}(\boldsymbol{x}) = 0, & \text{if } \boldsymbol{x}[1] > 0, \\ \pi_{(1)}(\boldsymbol{x}) = \frac{2}{3}, \pi_{(2)}(\boldsymbol{x}) = \frac{1}{3}, & \pi_{(3)}(\boldsymbol{x}) = \pi_{(4)}(\boldsymbol{x}) = 0, & \text{if } \boldsymbol{x}[1] < 0, \end{cases}$$

*and $\mathbb{P}(\boldsymbol{x}[1] > 0) = p_1$. Suppose the estimated probability of $\boldsymbol{x}$ provided by the model is:*

$$\begin{cases} \hat{\pi}_{(1)}(\boldsymbol{x}) = 1 - 3\xi \cdot \varepsilon, & \hat{\pi}_{(2)}(\boldsymbol{x}) = \hat{\pi}_{(3)}(\boldsymbol{x}) = \hat{\pi}_{(4)}(\boldsymbol{x}) = \xi \cdot \varepsilon, & \text{if } \boldsymbol{x}[1] > 0, \\ \hat{\pi}_{(1)}(\boldsymbol{x}) = \frac{2}{3} - \xi \cdot \varepsilon, \hat{\pi}_{(2)}(\boldsymbol{x}) = \frac{1}{3} - \xi \cdot \varepsilon, \hat{\pi}_{(3)}(\boldsymbol{x}) = \hat{\pi}_{(4)}(\boldsymbol{x}) = \xi \cdot \varepsilon, & \text{if } \boldsymbol{x}[1] < 0, \end{cases}$$

*where $\varepsilon \sim \text{Bernoulli}(p_2)$ and $\xi < \frac{1}{9}$.*

*Then the average prediction set size given by truncated CP is $(q_1' + q_2') + (q_3' + q_4' + q_5')[2 - p_1]$, which is smaller than $q_1' + q_2'[1 + p_2(1 - p_1)] + q_3'[2 - p_1] + q_4'[2 - p_1 + p_1 p_2] + q_5' \times [2 - p_1 + p_2 + p_1 p_2]$*

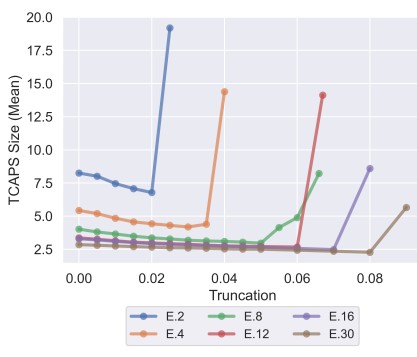

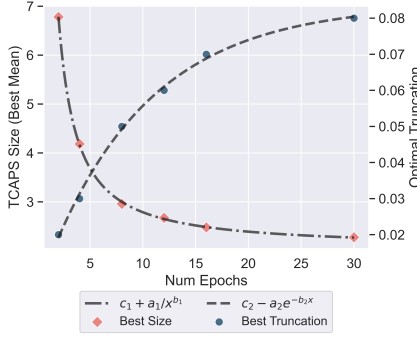

(a) Size vs. Truncation across Epochs

(b) Best Size & Truncation Dynamic

Figure 3: Relation between first-stage model quality (training epochs), truncation level, and prediction set size. **As first-stage model quality improves, the model better separates signal from noise (higher best truncation level), enabling truncation to achieve greater efficiency (reduced prediction set size),** with more details in Appendix C.3.

*without truncation when $k_{\mathrm{reg}} = 2$, where $q'_1, \ldots, q'_5 \in (0, 1)$ are probabilities, with their values and more details given in Appendix B.4*

To simulate group sparsity, we create two groups based on features with sparsity ratios of 25% and 75%, respectively. Intuitively, the ideal $k_{\mathrm{reg}}$ should correspond to the number of non-zero elements in the probability vector, effectively penalizing positions with small noises that should be estimated as zero. However, as the hyperparameter $k_{\mathrm{reg}}$ is fixed, it fails to adapt to the varying sparsity. Consequently, the prediction set becomes larger than the size of the oracle set predictor.

## 5 EXPERIMENTS

In this section, we aim to empirically demonstrate how incorporating a sparsity prior into the model using a truncation framework enhances the efficiency of the CP classification task. To achieve this, we conduct ablation studies on the truncation level, sparsity ratio, and the first-stage estimator using both synthetic (Section 5.1) and real-world datasets (Section 5.2).

### 5.1 SIMULATION DATA

**Data Generating Procedure.** To simulate group sparsity structure (Definition 2), we design a classification dataset with a hierarchical structure. Specifically, we denote the label classes as $\{y_i^{(j)} : j \in \{1, \ldots, C\}, i \in \{1, \ldots, N_j\}\}$, where the superscript $j$ represents the $j$-th large group, the subscript $i$ indicates the $i$-th small class within the $j$-th large group, $C$ denotes the number of large groups, and $N_j$ represents the number of classes in the $j$-th group. Given a data instance $\boldsymbol{x} = (\boldsymbol{x}^1, \boldsymbol{x}^2)$, where $\boldsymbol{x}^1 \in \mathbb{R}^{p_1}$ and $\boldsymbol{x}^2 \in \mathbb{R}^{p_2}$ are the covariates which decide the label of large group and small class. The large group label of $\boldsymbol{x}$ is given by $y_j = \arg\max_{y_j} \exp[\mathbf{w}_j^\top \boldsymbol{x}^1 + b_j]/\sum_{t=1}^C \exp[\mathbf{w}_t^\top \boldsymbol{x}^1 + b_t]$, where $(\mathbf{w}_j^\top, b_j)$ are the parameters associated with label $y_j$ for $j \in \{1, \ldots, C\}$. Furthermore, the probability distribution of $\boldsymbol{x}$ within the group $y_j$ is given by:

$$P(Y = y_i^{(j)} \mid X = \boldsymbol{x}^2, Y^{(j)} = y_j) = \frac{\exp[\mathbf{w}_{(i,j)}^\top \boldsymbol{x}^2 + b_{(i,j)}]}{\sum_{t=1}^{N_j} \exp[\mathbf{w}_{(i,t)}^\top \boldsymbol{x}^2 + b_{(i,t)}]},$$

where $(\mathbf{w}_{(i,j)}^\top, b_{(i,j)})$ are the parameters of class $i$ within the $j$-th group, with $i \in \{1, \ldots, N_j\}$.

**Setup.** We generate 200 samples per small class, yielding a total sample size of $200 \times (\sum_{j=1}^C N_j)$, and set $p_1 = 20$ and $p_2 = 30$. To simulate different sparsity ratios, the group size $C \in \{3, 6, 9\}$ and small classes $N_j \in \{10, 20, 30, 40, 50, 70\}$. Specific settings are denoted by `GroupSize` and `Num`.[3] The analysis focuses on three methods: APS (`APS`), APS with truncation (`TCAPS`), and RAPS (`RAPS`). Additional ablation studies on various CP methods are presented in Appendix D.2.

---

[3] `GroupSize` = [10,20,30] and `Num`=2-2-2 mean $C = 6$, with 2 groups for each class size $N_j = 10, 20, 30$.

Table 1: Coverage and prediction set size for APS, RAPS, and TCAPS on simulation data under varying sparsity ratios. The detailed results and the group coverage comparison are in Table 8, 9.

| Group | | APS | | RAPS | | TCAPS | |
|---|---|---|---|---|---|---|---|
| GSize | GNum | Coverage | Size | Coverage | Size | Coverage | Size |
| [10,20,30] | 1-1-1 | 90.71±0.01 | 5.43±0.10 | 90.42±0.01 | 4.76±0.32 | 90.54±0.01 | **4.70**±0.05 |
| | 2-2-2 | 91.46±0.01 | 6.42±0.05 | 90.46±0.01 | 4.41±0.09 | 90.47±0.01 | **4.34**±0.06 |
| | 3-3-3 | 91.57±0.01 | 5.49±0.05 | 90.37±0.03 | 3.51±0.07 | 90.50±0.01 | **3.45**±0.04 |

Table 2: Coverage and prediction set size for APS, RAPS, and TCAPS on Imagenet-Val. The group coverage comparison is provided in Table 10.

| Method | APS | | RAPS | | TCAPS | |
|---|---|---|---|---|---|---|
| Model | Coverage | Size | Coverage | Size | Coverage | Size |
| ResNeXt101 | 93.81±0.01 | 20.16±1.03 | 90.76±0.01 | 2.06±0.04 | 90.99±0.01 | **2.03**±0.03 |
| ResNet152 | 93.66±0.01 | 10.45±0.55 | 90.69±0.01 | 2.19±0.10 | 90.67±0.01 | **2.13**±0.05 |
| ResNet101 | 93.60±0.01 | 10.98±0.43 | 90.56±0.01 | 2.42±0.12 | 90.86±0.01 | **2.23**±0.03 |
| ResNet50 | 93.42±0.01 | 12.56±0.73 | 90.52±0.01 | 2.62±0.01 | 90.56±0.01 | **2.41**±0.03 |
| ResNet18 | 92.35±0.01 | 16.15±1.02 | 90.14±0.01 | 4.85±0.65 | 90.56±0.01 | **4.63**±0.10 |
| DenseNet161 | 93.57±0.01 | 12.65±0.24 | 90.71±0.01 | 2.41±0.04 | 90.40±0.01 | **2.22**±0.03 |
| VGG16 | 92.88±0.01 | 14.32±0.87 | 90.35±0.01 | 3.72±0.15 | 90.39±0.01 | **3.41**±0.05 |
| Inception | 92.57±0.01 | 89.85±2.98 | 90.28±0.01 | 5.47±0.25 | 90.35±0.01 | **5.16**±0.13 |
| ShuffleNet | 92.89±0.01 | 32.35±1.10 | 90.23±0.01 | 5.52±0.78 | 90.34±0.01 | **4.77**±0.08 |

**Ablation Study on the Truncation Level.** As shown in fig. 2, both size and coverage curves exhibit elbow points as the truncation level increases. Specifically, in the descending region, a higher truncation level leads to a sharp reduction in the output size, with a slight decrease in group coverage. This behavior highlights an optimal balance between two factors under the coverage constraint.

**Ablation Study on the Sparsity Ratio.** We also analyze how the sparsity ratio influences the size of prediction sets in Table 1. As the Num increases from 1 to 3, the setting becomes more sparse, leading to a greater increase in size compared to the vanilla APS. This demonstrates that higher sparsity ratios contribute to a larger gain in the prediction set size.

**Ablation Study on the First-Stage Estimator.** As shown in fig. 3, the elbow truncation position increases as the number of training epochs grows, while the corresponding size gradually decreases. Specifically, as the training epochs increase, the first-stage estimator becomes more accurate, making the gap between noise and signal more pronounced, thereby providing more room for truncation.

## 5.2 Real Data

In this section, we present experiments on real data to evaluate the effect of the truncation framework. Specifically, we assessed the performance of different CP methods (APS, TCAPS, RAPS) on various pre-trained models using the ImageNet-Val dataset (Deng et al., 2009). To further demonstrate the general applicability of the truncation framework, we apply it to the medical image data on PathMNIST (Kather et al., 2019), OrganAMNIST (Bilic et al., 2023), TissueMNIST (Woloshuk et al., 2021), BloodMNIST (Acevedo et al., 2020), DermaMNIST (Tschandl et al., 2018) and ChestXray (Wang et al., 2017), with details in Appendix D.3.

As illustrated in Table 2, 10, compared to APS, TCAPS truncates labels with low estimated probability values, fully leveraging the sparse prior to significantly shorten prediction set size without significantly compromising group coverage. Therefore, compared to RAPS, TCAPS further enhances efficiency. Additional ablation studies on base classifier quality (Appendix D.4) demonstrate that the benefits of truncation are robust to variations in classifier quality.

## 6 Conclusion

This paper proposes a truncation framework to incorporate the sparsity prior into the CP algorithm for classifiers. Intuitively, the estimated probability vector consists of two components: the signal and small noise, shaped by the model's softmax layer. By selecting the truncation level to separate signal from noise, the method suppresses small noise and amplifies the variance among signals. The truncation is post-hoc and orthogonal to score function design, ensuring compatibility with existing approaches, including RAPS. For further exploration, while classification tasks are widely applicable, incorporating prior information into regression models or other data structures might enhance the efficiency of CP algorithms in broader contexts.

ETHICS STATEMENT

This research is methodological, focusing on the development of a truncation–normalization framework that integrates sparsity priors into conformal prediction for classification tasks to enhance efficiency. The study does not involve human subjects and therefore did not require Institutional Review Board (IRB) approval. All experiments were performed on standard, publicly available benchmarks widely used in the machine learning community. No new data were collected, and no personally identifiable or sensitive information was processed, thereby avoiding concerns related to data privacy and security.

REPRODUCIBILITY STATEMENT

To ensure reproducibility, we provide comprehensive descriptions of both our theoretical results and experimental setup. The theoretical contributions in Section 4 are supported by complete mathematical proofs in Appendix B. Details of the experimental setup are provided in Section 5, along with additional implementation information, including data splitting protocols, the choice of truncation level, and computing devices, which are presented in Appendix C and Appendix D. The source code will be released publicly upon publication.

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

# Appendix

In this section, we first summarize the main contributions of this paper and discuss related work in Appendix A. Theoretical justifications and computational details are provided in Appendix B. Practical guidance on selecting the truncation level, along with key factors influencing this choice, is presented in Appendix C. Additional experiments on both simulation and real data are detailed in Appendix D. Finally, the discussion regarding the use of large language models in this paper is provided in Appendix E.

## A    SUMMARIZED MAIN CONTRIBUTIONS AND RELATED WORKS

In this section, we first summarize the main contributions of this paper in Appendix A.1. Furthermore, the detailed descriptions of three baseline conformal prediction methods for classification tasks are provided in Appendix A.2. Finally, we present an additional review on the use of sparsity priors in statistics and machine learning in Appendix A.3.

### A.1    SUMMARIZED MAIN CONTRIBUTIONS

Our contributions are mainly three-fold:

- We propose a truncation framework (Algorithm 1) that incorporates sparsity prior information into the CP algorithm. It is simple yet effective, and preserves the post-hoc nature of CP. Moreover, it is applicable across various classification CP methods.

- We theoretically demonstrate a U-shaped relationship between the truncation level and the prediction set size in Theorem 4. As the truncation level increases, the prediction set size first decreases and then increases. This finding provides a crucial practical guide (Corollary 5): by selecting the truncation level at the "elbow" of this curve, the method effectively separates signal from noise. This proper selection ensures improved efficiency and more compact prediction sets.

- Extensive experiments on both synthetic and real-world datasets (Section 5) demonstrate that, incorporating the sparsity prior could enhance the efficiency of the CP algorithm by choosing a truncation level near the "elbow" position. The results (fig. 3) further reveal that the "elbow" position shifts upward with training epochs as the improved model quality widens the gap between signal and noise, enabling more effective truncation.

### A.2    DETAILS OF CONFORMAL PREDICTION IN CLASSIFICATION

In this section, we provide the details of three baseline methods of conformal predictions in classification tasks. The core of these methods is the score function, a critical element that significantly influences how well the CP procedure adapts and its overall efficiency.

**Least Ambiguous Set-valued Classifier (LAC, Sadinle et al. (2019)).** In LAC, the score function is defined as $s(\boldsymbol{x}, y; \hat{\pi}) = 1 - \hat{\pi}_y(\boldsymbol{x})$, aiming to approximate the oracle solution of the optimization problem that minimizes the size of the set predictor while maintaining marginal coverage guarantees: $\min_{H:\mathcal{X}\to 2^{\mathcal{Y}}} \mathbb{E}|H(\boldsymbol{X})|$ subject to $\mathbb{P}(Y \notin H(\boldsymbol{X})) \le \alpha$. LAC achieves the smallest possible average set size under the strong assumption that $\hat{\pi}(\boldsymbol{x})$ perfectly represents the true conditional probability. But this assumption is unrealistic in practical scenarios, especially when data are limited. Moreover, LAC's conditional coverage guarantees are also restricted.

**Adaptive Prediction Sets (APS, Romano et al. (2020a)).** The conditional coverage is improved by APS with the score

$$s(\boldsymbol{x}, y; \hat{\pi}) = \sum_{y'=1}^{K} \hat{\pi}_{y'}(\boldsymbol{x}) \mathbb{I}_{\{\hat{\pi}_{y'}(\boldsymbol{x}) \ge \hat{\pi}_y(\boldsymbol{x})\}}. \tag{4}$$

It is an approximation to the objective of minimizing efficiency while ensuring conditional coverage:

$$\min_{H:\mathcal{X}\to 2^{\mathcal{Y}}} \mathbb{E}|H(\boldsymbol{X})| \text{ subject to } \mathbb{P}(Y \notin H(\boldsymbol{X}) \mid \boldsymbol{X} = \boldsymbol{x}) \le \alpha \text{ for all } \boldsymbol{x} \in \mathcal{X}. \tag{5}$$

The oracle set predictor is defined as the solution to equation 5 when the

$$\mathcal{C}^{\text{oracle}}(\boldsymbol{x}; f = \pi, \tau = 1 - \alpha) = \{ \text{`}y\text{'} \text{ indices of the } L(\boldsymbol{x}; \pi, 1 - \alpha) \text{ largest } \pi_y(\boldsymbol{x}) \} , \tag{6}$$

where $L(\boldsymbol{x}; \pi, \tau) = \min \left\{ c \in \{1, \dots, K\} : \sum_{i=1}^{c} \pi_{(i)}(\boldsymbol{x}) \geq \tau \right\}$.

However, the efficiency of APS is often affected by small values in the estimated probability vector. Achieving optimal performance requires knowledge of the joint distribution, which might be unavailable in practice.

**Regularized Adaptive Prediction Sets (RAPS, Angelopoulos et al. (2020))**. To improve the efficiency of APS, RAPS defines the score as $s(\boldsymbol{x}, y; \hat{\pi}) = \sum_{y'=1}^{K} \hat{\pi}_{y'}(\boldsymbol{x}) \mathbb{I}_{\{\hat{\pi}_{y'}(\boldsymbol{x}) \geq \hat{\pi}_y(\boldsymbol{x})\}} + \gamma (o_{\boldsymbol{x}}(y) - k_{\text{reg}})_+$, where $o_{\boldsymbol{x}}(y)$ is the index to which label $y$ is mapped after the scores are sorted, and $(\cdot)_+ := \max\{\cdot, 0\}$. The hyperparameters $\gamma$ and $k_{\text{reg}}$ control the intensity of the penalty. RAPS addresses the issue introduced by small noises in the estimated probability vector by penalizing difficult examples, however, it is less adaptive to the feature, which may reduce efficiency when data exhibit group sparsity (Example 11).

### A.3 RELATED WORKS ON SPARSITY PRIOR

Sparsity priors play a central role in high-dimensional statistics by imposing low-dimensional structures that render otherwise intractable problems more manageable. For high-dimensional vector estimation, a representative example is the Lasso (Tibshirani, 1996), which introduces an $L_1$ penalty to simultaneously perform variable selection and regularization by shrinking unimportant coefficients to zero, thereby producing sparse and interpretable models. In the context of high-dimensional covariance estimation, sparsity assumptions are often imposed on the precision matrix (the inverse covariance matrix), which is crucial in graphical modeling. The Graphical Lasso (Friedman et al., 2008; Meinshausen & Bühlmann, 2006) extends the $L_1$-regularization framework to this setting, allowing one to estimate sparse precision matrices and uncover conditional dependencies between variables in Gaussian graphical models.

Beyond estimation problems, sparsity priors have also been explored in classification tasks. The standard Softmax function assigns strictly positive probabilities to all classes, which may reduce interpretability in high-dimensional settings with many classes. To mitigate this limitation, alternative activation functions such as Sparsemax (Martins & Astudillo, 2016) and its variants (Peters et al., 2019; Blondel et al., 2020) have been developed. These methods yield sparse probability distributions with exact zeros, enabling models to produce more decisive, interpretable predictions and improving scalability when dealing with a large number of output categories. However, incorporating such sparsity-aware activation functions into conformal prediction typically requires retraining the base model, thereby undermining the post-hoc nature that is central to CP.

## B THEORETICAL JUSTIFICATIONS AND COMPUTATIONAL DETAILS

In this section, we present theoretical justifications and computational details that support our main results in Section 4. We start with the proof of Theorem 4 in Appendix B.1, which explains why the size of the resulting prediction set first decreases and then increases as the truncation level grows. Appendix B.2 establishes the oracle property of the truncated conformal method by proving Theorem 7, showing that under a consistent first-stage classifier, the truncated conformal set asymptotically recovers the oracle predictor. Besides the asymptotic oracle recovery, the proof of the standard marginal coverage guarantee of truncated CP (Theorem 8) is provided in Appendix B.3. Finally, the detailed computations of the prediction set size in Appendix B.4 are provided in Section 4.4.

### B.1 PROOF OF THEOREM 4

The proof of Theorem 4 relies on two auxiliary lemmas. The first characterizes a key property of the gradient of the gap function, and the second provides an upper bound on the difference between two gap function values computed on the same sample but with different summation indices.

**Lemma 12.** *For any $\boldsymbol{x} \in \mathcal{X}$, let $\hat{\pi}(\boldsymbol{x}) = \sigma(\boldsymbol{z}(\boldsymbol{x})) \in \Delta^{K-1}$ represent the estimated probability, where $\boldsymbol{z}$ is the logit vector output by a base model and $\sigma(\cdot)$ denotes the softmax operator. Denote*

$\Delta \boldsymbol{z} = \boldsymbol{z}[1] - \boldsymbol{z}[2]$ *as the difference between the top 1 and top 2 logits. Assume that there exists* $\gamma$ *such that* $\hat{\pi}_y(\boldsymbol{x}) > \gamma$ *for any sample* $(\boldsymbol{x}, y)$. *If choose* $\lambda \le \gamma$, *for* $M \le L_\lambda$, *it holds that:*

$$\begin{cases} \text{If} & \lambda < \frac{1}{K}, \exp(\Delta \boldsymbol{z}) > (K-1)(1 - K\lambda) \implies \nabla_{\boldsymbol{z}[1]} g(\boldsymbol{z}; \lambda, M) > 0, \\ \text{If} & \lambda \ge \frac{1}{K} \implies \nabla_{\boldsymbol{z}[1]} g(\boldsymbol{z}; \lambda, M) > 0. \end{cases}$$

*Proof.* For $\lambda \le \gamma$, since $\hat{\pi}_y(\boldsymbol{x}) > \gamma$, it holds that $S_\lambda(\boldsymbol{x}) \ge 1$ and $L_\lambda \le S_\lambda(\boldsymbol{x})$. Consider the derivative of the following exponential term with respect to $\boldsymbol{z}[1]$:

$$\frac{\partial \left( \exp(\boldsymbol{z}[j]) \mathbb{I}_{\{\exp(\boldsymbol{z}[j]) > \lambda \sum_{i=1}^{K} \exp(\boldsymbol{z}[i])\}} \right)}{\partial \boldsymbol{z}[1]}$$

$$= \frac{\partial (\exp(\boldsymbol{z}[j]))}{\partial \boldsymbol{z}[1]} \mathbb{I}_{\{\exp(\boldsymbol{z}[j]) > \lambda \sum_{i=1}^{K} \exp(\boldsymbol{z}[i])\}} + \frac{\partial \left( \mathbb{I}_{\{\exp(\boldsymbol{z}[j]) > \lambda \sum_{i=1}^{K} \exp(\boldsymbol{z}[i])\}} \right)}{\partial \boldsymbol{z}[1]} \exp(\boldsymbol{z}[j])$$

$$= \begin{cases} \exp(\boldsymbol{z}[1]), & \text{if} \quad j = 1, \\ 0, & \text{others .} \end{cases}$$

For any $M \le L_\lambda \le S_\lambda(\boldsymbol{x})$, the gap function is defined as:

$$g(\boldsymbol{z}; \lambda, M) = \sum_{i=1}^{M} \hat{\pi}_{(i)}(\boldsymbol{x}) - \hat{\pi}_{(i)}^{\lambda}(\boldsymbol{x})$$

$$= \sum_{i=1}^{M} \hat{\pi}_{(i)}(\boldsymbol{x}) - \frac{\sum_{i=1}^{M} \hat{\pi}_{(i)}(\boldsymbol{x})}{\sum_{i=1}^{S_\lambda(\boldsymbol{x})} \hat{\pi}_{(i)}(\boldsymbol{x})}$$

$$= \frac{\sum_{i=1}^{M} \exp(\boldsymbol{z}[i])}{\sum_{i=1}^{K} \exp(\boldsymbol{z}[i])} - \frac{\sum_{i=1}^{M} \exp(\boldsymbol{z}[i])}{\sum_{i=1}^{S_\lambda(\boldsymbol{x})} \exp(\boldsymbol{z}[i])}.$$

Then compute the differentiate with respect to $\boldsymbol{z}[1]$ as:

$$\nabla_{\boldsymbol{z}[1]} g(\boldsymbol{z}; \lambda, M) = \exp(\boldsymbol{z}[1]) \left[ \frac{\sum_{i=M+1}^{K} \exp(\boldsymbol{z}[i])}{(\sum_{i=1}^{K} \exp(\boldsymbol{z}[i]))^2} - \frac{\sum_{i=M+1}^{S_\lambda(\boldsymbol{x})} \exp(\boldsymbol{z}[i])}{(\sum_{i=1}^{S_\lambda(\boldsymbol{x})} \exp(\boldsymbol{z}[i]))^2} \right]$$

$$= \frac{\exp(\boldsymbol{z}[1]) \sum_{i=S_\lambda(\boldsymbol{x})+1}^{K} \exp(\boldsymbol{z}[i])}{(\sum_{i=1}^{K} \exp(\boldsymbol{z}[i]))^2 (\sum_{i=1}^{S_\lambda(\boldsymbol{x})} \exp(\boldsymbol{z}[i]))^2} \times$$

$$\left[ \left( \sum_{i=1}^{M} \exp(\boldsymbol{z}[i]) \right)^2 - \left( \sum_{i=M+1}^{S_\lambda(\boldsymbol{x})} \exp(\boldsymbol{z}[i]) \right) \left( \sum_{i=S_\lambda(\boldsymbol{x})+1}^{K} \exp(\boldsymbol{z}[i]) \right) - \left( \sum_{i=M+1}^{S_\lambda(\boldsymbol{x})} \exp(\boldsymbol{z}[i]) \right)^2 \right],$$

note that if $M = S_\lambda(\boldsymbol{x})$ then $\sum_{i=M+1}^{S_\lambda(\boldsymbol{x})} \exp(\boldsymbol{z}[i]) = 0$ .

Therefore, $\nabla_{\boldsymbol{z}[1]} g(\boldsymbol{z}; \lambda, M) > 0$ holds if and only if the following inequality holds:

$$\left( \sum_{i=1}^{M} \hat{\pi}_{(i)}(\boldsymbol{x}) \right)^2 - \left( \sum_{i=M+1}^{S_\lambda(\boldsymbol{x})} \hat{\pi}_{(i)}(\boldsymbol{x}) \right) \left( \sum_{i=S_\lambda(\boldsymbol{x})+1}^{K} \hat{\pi}_{(i)}(\boldsymbol{x}) \right) - \left( \sum_{i=M+1}^{S_\lambda(\boldsymbol{x})} \hat{\pi}_{(i)}(\boldsymbol{x}) \right)^2 > 0. \quad (7)$$

Since $\sum_{i=S_\lambda(\boldsymbol{x})+1}^{K} \hat{\pi}_{(i)}(\boldsymbol{x}) < K\lambda$ and $\sum_{i=1}^{K} \hat{\pi}_{(i)}(\boldsymbol{x}) = 1$, the upper bound of the right-hand side of (7) is given by:

$$\left( \sum_{i=M+1}^{S_\lambda(\boldsymbol{x})} \hat{\pi}_{(i)}(\boldsymbol{x}) \right) \left( \sum_{i=S_\lambda(\boldsymbol{x})+1}^{K} \hat{\pi}_{(i)}(\boldsymbol{x}) \right) + \left( \sum_{i=M+1}^{S_\lambda(\boldsymbol{x})} \hat{\pi}_{(i)}(\boldsymbol{x}) \right)^2$$

$$< \left( 1 - \sum_{i=1}^{M} \hat{\pi}_{(i)}(\boldsymbol{x}) \right) \left( 1 - \sum_{i=1}^{M} \hat{\pi}_{(i)}(\boldsymbol{x}) - K\lambda \right).$$

$(8)$

Combining (7) and equation 8 yields a sufficient condition for the positivity of the gradient:

$$\left(\sum_{i=1}^{M}\hat{\pi}_{(i)}(\boldsymbol{x})\right)^2 \geq \left(1 - \sum_{i=1}^{M}\hat{\pi}_{(i)}(\boldsymbol{x})\right)\left(1 - \sum_{i=1}^{M}\hat{\pi}_{(i)}(\boldsymbol{x}) - K\lambda\right) \implies \nabla_{\boldsymbol{z}[1]}g(\boldsymbol{z};\lambda,M) > 0. \quad (9)$$

Solving the inequality in equation 9, it holds that:

$$\begin{cases} \text{If} \quad \lambda < \frac{1}{K}, \quad \sum_{i=1}^{M}\hat{\pi}_{(i)}(\boldsymbol{x}) > 1 + \frac{1}{K\lambda - 2} \implies \nabla_{\boldsymbol{z}[1]}g(\boldsymbol{z};\lambda,M) > 0, \\ \text{If} \quad \lambda \geq \frac{1}{K} \implies \nabla_{\boldsymbol{z}[1]}g(\boldsymbol{z};\lambda,M) > 0. \end{cases} \quad (10)$$

Finally, note that $\sum_{i=1}^{K}\exp(\boldsymbol{z}[i]) \leq \exp(\boldsymbol{z}[1]) + (K-1)\exp(\boldsymbol{z}[2])$, so we obtain the following lower bound:

$$\sum_{i=1}^{M}\hat{\pi}_{(i)}(\boldsymbol{x}) \geq \hat{\pi}_{(1)}(\boldsymbol{x}) \geq \frac{1}{1 + (K-2)\exp(-\Delta\boldsymbol{z})}. \quad (11)$$

Combine equation 10 and equation 11, the conclusion holds. $\qquad\square$

**Lemma 13.** *Define the probability difference before and after truncation as $d(\boldsymbol{x};\lambda,i) = \hat{\pi}_{(i)}(\boldsymbol{x}) - \hat{\pi}_{(i)}^{\lambda}(\boldsymbol{x})$. The gap function is then defined by $g(\boldsymbol{z};\lambda,M) = \sum_{i=1}^{M}[\sigma_i(\boldsymbol{z}) - T_\lambda(\sigma(\boldsymbol{z}))_i] = \sum_{i=1}^{M}d(\boldsymbol{x};\lambda,i)$. Then, it holds that:*

$$\forall M, L^q \in \{1, \cdots, K\}, \quad |g(\boldsymbol{z};\lambda,M) - g(\boldsymbol{z};\lambda,L^q)| \leq \left|\sum_{i=1}^{S_\lambda(\boldsymbol{x})}d(\boldsymbol{x};\lambda,i)\right|. \quad (12)$$

*Proof.* Without loss of generality, assume $M > L^q$. Then, the absolute difference in equation 12 can be computed as:

$$|g(\boldsymbol{z};\lambda,M) - g(\boldsymbol{z};\lambda,L^q)| = \left|\sum_{i=1}^{M}d(\boldsymbol{x};\lambda,i) - \sum_{i=1}^{L^q}d(\boldsymbol{x};\lambda,i)\right| = \left|\sum_{i=L^q+1}^{M}d(\boldsymbol{x};\lambda,i)\right|.$$

Recall that for all $i \leq S_\lambda(\boldsymbol{x})$, $d(\boldsymbol{x};\lambda,i) \leq 0$; while for all $i > S_\lambda(\boldsymbol{x})$, $d(\boldsymbol{x};\lambda,i) \geq 0$. Based on the position of $L^q + 1$ and $M$ relative to $S_\lambda(\boldsymbol{x})$, we consider the following three cases:

- **Case 1:** $L^q + 1 \geq S_\lambda(\boldsymbol{x})$. In this case, the summation includes only non-negative differences. Hence, the maximum is achieved by summing all positive differences:

$$\left|\sum_{i=L^q+1}^{M}d(\boldsymbol{x};\lambda,i)\right| \leq \left|\sum_{i=S_\lambda(\boldsymbol{x})+1}^{K}d(\boldsymbol{x};\lambda,i)\right|.$$

- **Case 2:** $M \leq S_\lambda(\boldsymbol{x})$. The summation includes only non-positive differences. Since $L^q + 1 \geq 2$, the following inequality holds:

$$\left|\sum_{i=L^q+1}^{M}d(\boldsymbol{x};\lambda,i)\right| \leq \left|\sum_{i=2}^{S_\lambda(\boldsymbol{x})}d(\boldsymbol{x};\lambda,i)\right|.$$

- **Case 3:** $L^q + 1 < S_\lambda(\boldsymbol{x}) < M$. The summation includes both positive and negative differences. In this case, the absolute sum can be bounded by the maximum of the previous two cases:

$$\left|\sum_{i=L^q+1}^{M}d(\boldsymbol{x};\lambda,i)\right| \leq \max\left\{\left|\sum_{i=S_\lambda(\boldsymbol{x})+1}^{K}d(\boldsymbol{x};\lambda,i)\right|, \left|\sum_{i=2}^{S_\lambda(\boldsymbol{x})}d(\boldsymbol{x};\lambda,i)\right|\right\}.$$

Note that by construction, the total sum of differences is zero, *i.e.*, $\sum_{i=1}^{K} d(\boldsymbol{x}; \lambda, i) = 0$, which implies

$$\sum_{i=1}^{S_\lambda(\boldsymbol{x})} d(\boldsymbol{x}; \lambda, i) = - \sum_{i=S_\lambda(\boldsymbol{x})+1}^{K} d(\boldsymbol{x}; \lambda, i),$$

and hence

$$\left| \sum_{i=1}^{S_\lambda(\boldsymbol{x})} d(\boldsymbol{x}; \lambda, i) \right| = \left| \sum_{i=S_\lambda(\boldsymbol{x})+1}^{K} d(\boldsymbol{x}; \lambda, i) \right|.$$

Since both partial sums are composed of terms with the same sign, removing any term strictly reduces the total magnitude. Therefore,

$$\left| \sum_{i=2}^{S_\lambda(\boldsymbol{x})} d(\boldsymbol{x}; \lambda, i) \right| < \left| \sum_{i=S_\lambda(\boldsymbol{x})+1}^{K} d(\boldsymbol{x}; \lambda, i) \right|,$$

which implies

$$\max \left\{ \left| \sum_{i=S_\lambda(\boldsymbol{x})+1}^{K} d(\boldsymbol{x}; \lambda, i) \right|, \left| \sum_{i=2}^{S_\lambda(\boldsymbol{x})} d(\boldsymbol{x}; \lambda, i) \right| \right\} = \left| \sum_{i=1}^{S_\lambda(\boldsymbol{x})} d(\boldsymbol{x}; \lambda, i) \right|.$$

Combining all cases together, it concludes that:

$$\forall M, L^q \in \{1, \ldots, K\}, \quad |g(\boldsymbol{z}; \lambda, M) - g(\boldsymbol{z}; \lambda, L^q)| \leq \left| \sum_{i=1}^{S_\lambda(\boldsymbol{x})} d(\boldsymbol{x}; \lambda, i) \right|.$$

$\square$

**Theorem 4** (Impact of Truncation Level on Size). *For any $\boldsymbol{x} \in \mathcal{X}$, let $\hat{\pi}(\boldsymbol{x}) = \sigma(\boldsymbol{z}(\boldsymbol{x})) \in \Delta^{K-1}$ represent the estimated probability, where $\boldsymbol{z}$ is the logit vector output by a base model. For a truncation level $\lambda > 0$, the sparsified probability vector for $\boldsymbol{x}$ is defined as $\hat{\pi}^\lambda(\boldsymbol{x}) = T_\lambda(\hat{\pi}(\boldsymbol{x}))$, as given in eq. (2). Let $\hat{q}^\lambda$ denote the quantile of the score computed on the calibration set. The size of the prediction set for $\boldsymbol{x}$ is given by $L_\lambda = L(\boldsymbol{x}; \hat{\pi}^\lambda, \hat{q}^\lambda) = \min \left\{ c \in \{1, \ldots, K\} : \sum_{i=1}^{c} \hat{\pi}_{(i)}^\lambda(\boldsymbol{x}) \geq \hat{q}^\lambda \right\}$. The number of non-zero elements in $\hat{\pi}^\lambda(\boldsymbol{x})$ is $S_\lambda(\boldsymbol{x}) = \max_i \{i : \hat{\pi}_y(\boldsymbol{x}) > \min(\lambda, \hat{\pi}_{(2)}(\boldsymbol{x}))\}$. The gap function is defined as $g(\boldsymbol{z}; \lambda, M) = \sum_{i=1}^{M} [\sigma_i(\boldsymbol{z}) - T_\lambda(\sigma(\boldsymbol{z}))_i]$. Let $\boldsymbol{z}[-1] = (\boldsymbol{z}[2], \ldots, \boldsymbol{z}[K])$ and $\Delta \boldsymbol{z} = \boldsymbol{z}[1] - \boldsymbol{z}[2]$. Under the following assumptions:*
*(C1) There exists $\delta$ such that $\hat{\pi}_y(\boldsymbol{x}) > \delta$ for any sample $(\boldsymbol{x}, y)$.*
*(C2) $\hat{q}^0$ and $\hat{q}^\lambda$ are derived from the same sample $\boldsymbol{x}^q$ in the calibration set.*
*(C3) $\sum_{i=S_\lambda(\boldsymbol{x}^q)+1}^{K} \hat{\pi}_{(i)}(\boldsymbol{x}^q) = o(\gamma)$.*
*(C4) $\boldsymbol{z}^q[1] > \boldsymbol{z}[1]$, where $\boldsymbol{z}^q$ is the logit vector of $\boldsymbol{x}^q$. For any $\boldsymbol{z}' = \theta \boldsymbol{z}^q[1] + (1-\theta)\boldsymbol{z}[1]$ ($\theta \in [0, 1]$) and $M \leq L_\lambda$, $|\nabla g_{\boldsymbol{z}[1]}(\boldsymbol{z}'; \lambda, M)(\boldsymbol{z}^q[1] - \boldsymbol{z}[1])| > |\nabla g_{\boldsymbol{z}[-1]}(\boldsymbol{z}'; \lambda, M)^\top (\boldsymbol{z}^q[-1] - \boldsymbol{z}[-1])|$.*
*(C5) The quantity $\Delta \boldsymbol{z}$ increases monotonically with the score $s(\boldsymbol{x}, y; \hat{\pi})$.*
*Then, for any truncation level $\lambda \leq \delta$, it holds that:*

$$\begin{cases} \text{If} \quad \lambda < \frac{1}{K}, \exp(\Delta \boldsymbol{z}) > (K-1)(1 - K\lambda) \Longrightarrow L_0 \geq L_\lambda, \\ \text{If} \quad \lambda \geq \frac{1}{K} \Longrightarrow L_0 \geq L_\lambda. \end{cases}$$

*Proof.* Let $\lambda \leq \gamma$, then $S_\lambda(\boldsymbol{x}) \geq 1$ and $L_\lambda \leq S_\lambda(\boldsymbol{x})$. By Lemma 12, the following holds:

$$\begin{cases} \text{If } \lambda < \frac{1}{K}, \quad \exp(\Delta \boldsymbol{z}) > (K-1)(1 - K\lambda) \Longrightarrow \nabla_{\boldsymbol{z}[1]} g(\boldsymbol{z}; \lambda, M) > 0, \\ \text{If } \lambda \geq \frac{1}{K} \Longrightarrow \nabla_{\boldsymbol{z}[1]} g(\boldsymbol{z}; \lambda, M) > 0. \end{cases}$$

From condition (C5), for any $\boldsymbol{x}'$ such that $s(\boldsymbol{x}, y) \leq s(\boldsymbol{x}', y) \leq s(\boldsymbol{x}^q, y)$, the inequality $\nabla_{\boldsymbol{z}[1]} g(\boldsymbol{z}; \lambda, M) > 0$ can be obtained via Lemma 12. Therefore, by condition (C4), it holds that:

$$g(\boldsymbol{z}^q, \lambda, M) - g(\boldsymbol{z}, \lambda, M) = \nabla_{\boldsymbol{z}[1]} g(\boldsymbol{z}'; \lambda, M)(\boldsymbol{z}^q[1] - \boldsymbol{z}[1]) + \nabla_{\boldsymbol{z}[-1]} g(\boldsymbol{z}'; \lambda, M)^\top (\boldsymbol{z}^q[-1] - \boldsymbol{z}[-1])$$
$$> 0, \quad \text{for } M \leq L_\lambda.$$

$$(13)$$

Furthermore, by Lemma 13, the following inequality holds:

$$\forall M, L^q \in \{1, \ldots, K\}, \quad |g(\boldsymbol{z}; \lambda, M) - g(\boldsymbol{z}; \lambda, L^q)| \leq \left| \sum_{i=1}^{S_\lambda(\boldsymbol{x})} d(\boldsymbol{x}; \lambda, i) \right|, \qquad (14)$$

where

$$d(\boldsymbol{x}; \lambda, i) = \hat{\pi}_{(i)}(\boldsymbol{x}) - \frac{\hat{\pi}_{(i)}(\boldsymbol{x})}{\sum_{j=1}^{S_\lambda(\boldsymbol{x})} \hat{\pi}_{(j)}(\boldsymbol{x})}, \quad i \leq S_\lambda(\boldsymbol{x}).$$

Since $\sum_{i=1}^{S_\lambda(\boldsymbol{x})} d(\boldsymbol{x}; \lambda; i) = \sum_{i=S_\lambda(\boldsymbol{x})+1}^{K} \hat{\pi}_{(i)}(\boldsymbol{x}) = o(\gamma)$, it follows from equation 13 and equation 14 that:

$$g(\boldsymbol{z}, \lambda, M) < g(\boldsymbol{z}^q, \lambda, L^q) + o(\gamma).$$

Moreover, from the definition of the gap function $g(\cdot; \cdot, \cdot)$ and condition (C2), the following inequality holds for all $M \in \{1, \ldots, L_\lambda\}$:

$$\sum_{i=1}^{M} \hat{\pi}_{(i)}(\boldsymbol{x}) - \hat{\pi}_{(i)}^{\lambda}(\boldsymbol{x}) < \hat{q}^0 - \hat{q}^\lambda + o(\gamma).$$

Now observe that for any $0 < t \leq \hat{q}^\lambda$, define

$$M_t := \min \left\{ l \in \{1, \ldots, K\} : \sum_{i=1}^{l} \hat{\pi}_{(i)}^{\lambda}(\boldsymbol{x}) \geq t \right\}.$$

Then $M_t \leq L_\lambda$. If $M_t > 1$, it holds that

$$\hat{q}^\lambda - \hat{q}^0 + \sum_{i=1}^{M_t - 1} \hat{\pi}_{(i)}(\boldsymbol{x}) - o(\gamma) \leq \sum_{i=1}^{M_t - 1} \hat{\pi}_{(i)}^{\lambda}(\boldsymbol{x}) < t.$$

This implies that

$$\min \left\{ l \in \{1, \ldots, K\} : \hat{q}_\lambda - \hat{q} + \sum_{i=1}^{l} \hat{\pi}_{(i)}(\boldsymbol{x}) - o(\gamma) \geq t \right\} \geq M_t.$$

That is,

$$\forall \, 0 < t \leq \hat{q}^\lambda : \quad \min \left\{ l : \sum_{i=1}^{l} \hat{\pi}_{(i)}^{\lambda}(\boldsymbol{x}) \geq t \right\} \leq \min \left\{ l : \hat{q}^\lambda - \hat{q}^0 + \sum_{i=1}^{l} \hat{\pi}_{(i)}(\boldsymbol{x}) - o(\gamma) \geq t \right\}.$$

In particular, choosing $t = \hat{q}^\lambda$, it holds that:

$$\min \left\{ l : \sum_{i=1}^{l} \hat{\pi}_{(i)}^{\lambda}(\boldsymbol{x}) \geq \hat{q}_\lambda \right\} \leq \min \left\{ l : \hat{q}^\lambda - \hat{q}^0 + \sum_{i=1}^{l} \hat{\pi}_{(i)}(\boldsymbol{x}) - o(\gamma) \geq \hat{q}^\lambda \right\}$$

$$\Longleftrightarrow \min \left\{ l : \sum_{i=1}^{l} \hat{\pi}_{(i)}^{\lambda}(\boldsymbol{x}) \geq \hat{q}^\lambda \right\} \leq \min \left\{ l : \sum_{i=1}^{l} \hat{\pi}_{(i)}(\boldsymbol{x}) \geq \hat{q}^0 + o(\gamma) \right\}.$$

This completes the proof and establishes the inequality $L_\lambda \leq L_0$.

$\square$

We further provide empirical validation to support the technical conditions in Theorem 4. To verify condition (C2), we compare the distributions of conformal scores before and after applying truncation, as shown in fig. 4. The analysis is conducted on simulated data with a group size configuration of 1-1-1, and the truncation level is set to 0.03. The results indicate that truncation preserves the relative ranking of class probabilities for most samples, resulting in minimal changes to the order

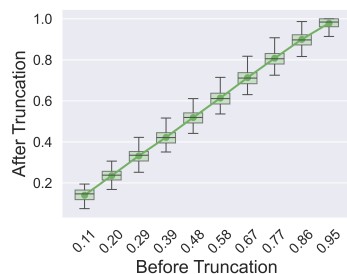
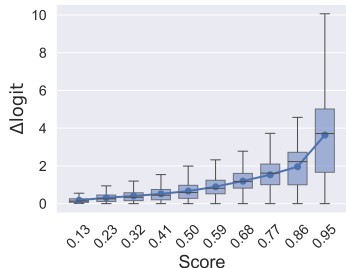

Figure 4: Conformal score with truncation probability sorted by conform score without truncation (low to high). This figure corresponds to assumption (C2).

Figure 5: Logit gap $\Delta z$ for each sample sorted by conformal score (low to high). This figure corresponds to assumptions (C4) and (C5).

of scores within the calibration set. This justifies condition (C2), which asserts that the quantiles $\hat{q}^0$ and $\hat{q}^\lambda$ are both derived from the same calibration sample $\boldsymbol{x}^q$.

Next, we investigate conditions (C4) and (C5). In fig. 5, we illustrate the relationship between the conformal score and the logit gap $\Delta z$ using the same simulated setup. The figure clearly supports condition (C5), as $\Delta z$ tends to increase with the score, reflecting a growing dominance of the largest logit $z_1$ over $z_2$, and, by extension, over $z_3, \ldots, z_K$. This supports the intuition behind condition (C4): namely, that the variation in the gap function $g(\boldsymbol{x}; \lambda, M)$ across different inputs is primarily driven by the magnitude of the first entry in their respective softmax vectors.

Finally, we emphasize that even if these assumptions are not strictly satisfied in practice, the macro-level U-shaped relationship between the prediction set size and truncation level given by Theorem 4 still emerges, as confirmed by the extensive experiments in Section 5. Future work will aim to establish theoretical guarantees under more relaxed conditions.

## B.2 PROOF OF THEOREM 7

In this section, we prove that the oracle set predictor could be asymptotically recovered after truncation, assuming the first-stage estimator is consistent based on the APS score function.

**Theorem 7** (Asymptotic Oracle Recovery). *For any $\boldsymbol{x} \in \mathcal{X}$, let $\pi(\boldsymbol{x})$ and $\hat{\pi}(\boldsymbol{x})$ denote the true and estimated probability simplex, where $\pi(\boldsymbol{x})$ has sparsity structure. Let the score function be the APS score defined in eq. (1). Denote the prediction set obtained by $\hat{\pi}(\cdot)$ after truncation with the level $\lambda$ as $\mathcal{C}_{\mathrm{TCAPS}}(\boldsymbol{x}; \hat{\pi}^\lambda, \hat{q}^\lambda) = \{y : s(\boldsymbol{x}, y; \hat{\pi}^\lambda) \leq \hat{q}^\lambda\}$, and the oracle prediction set based on $\pi(\cdot)$ as $\mathcal{C}^{\mathrm{oracle}}(\boldsymbol{x}; \pi, q) = \{y : s(\boldsymbol{x}, y; \pi) \leq q\}$, where $\hat{q}^\lambda$ and $q$ are the $\lceil(1-\alpha)(1+n)\rceil$-largest value of scores computed on the calibration set $\{(\boldsymbol{x}_i, y_i)\}_{i=1}^n$ using $\hat{\pi}^\lambda$ and $\pi$. Let $S_{\boldsymbol{x}} = \{i : \pi_i(\boldsymbol{x}) > 0\}$ be the support of $\pi(\boldsymbol{x})$, and $S_{\boldsymbol{x}}^C = \{i : \pi_i(\boldsymbol{x}) = 0\}$. Assume there exists $\delta$ such that $\min_{i \in S_{\boldsymbol{x}}} \hat{\pi}_i(\boldsymbol{x}) > \delta > \max_{i \in S_{\boldsymbol{x}}^c} \hat{\pi}_i(\boldsymbol{x})$ for all $\boldsymbol{x}$, and $\|\hat{\pi}(\boldsymbol{x}) - \pi(\boldsymbol{x})\|_2 \to 0$ as $n \to \infty$. Choosing $\lambda = \delta$ yields*

$$\mathcal{C}_{\mathrm{TCAPS}}(\boldsymbol{x}; \hat{\pi}^\lambda, \hat{q}^\lambda) \to \mathcal{C}^{\mathrm{oracle}}(\boldsymbol{x}; \pi, q) \text{ as } n \to \infty.$$

*In contrast, the prediction set $\mathcal{C}(\boldsymbol{x}; \hat{\pi}, \hat{q})$ constructed without truncation is larger than the oracle set, i.e., $|\mathcal{C}(\boldsymbol{x}; \hat{\pi}, \hat{q})| > |\mathcal{C}^{\mathrm{oracle}}(\boldsymbol{x}; \pi, q)|$ for any sample size $n$.*

*Proof.* Since $\|\hat{\pi}(\boldsymbol{x}) - \pi(\boldsymbol{x})\|_2 \leq \|\hat{\pi}(\boldsymbol{x}) - \pi(\boldsymbol{x})\|_1 \leq \sqrt{K}\|\hat{\pi}(\boldsymbol{x}) - \pi(\boldsymbol{x})\|_2$, it holds that

$$
\begin{aligned}
s(\boldsymbol{x}, y; \hat{\pi}) &= \sum_{y'=1}^{K} \hat{\pi}_{(i)}^{\lambda}(\boldsymbol{x}) \mathbb{I}_{\{\hat{\pi}_{(y')}(\boldsymbol{x}) \geq \hat{\pi}_y(\boldsymbol{x})\}} \\
&= \frac{\sum_{y'=1}^{K} \hat{\pi}_{y'}(\boldsymbol{x}) \mathbb{I}_{\{\hat{\pi}_{y'}(\boldsymbol{x}) > \hat{\pi}_y(\boldsymbol{x})\}} + \hat{\pi}_{(y)}(\boldsymbol{x})}{1 - \sum_{i \in S_{\boldsymbol{x}}^c} \hat{\pi}_i(\boldsymbol{x})} \\
&= \frac{\sum_{y'=1}^{K} \hat{\pi}_{y'}(\boldsymbol{x}) \mathbb{I}_{\{\hat{\pi}_{y'}(\boldsymbol{x}) > \hat{\pi}_y(\boldsymbol{x})\}} + \hat{\pi}_{(y)}(\boldsymbol{x})}{1 - |\hat{\pi}_{S_x^c}(\boldsymbol{x}) - \pi_{S_x^c(\boldsymbol{x})}|} \\
&\leq \frac{2\|\hat{\pi}(\boldsymbol{x}) - \pi(\boldsymbol{x})\|_1 + \sum_{y'=1}^{K} \pi_{y'}(\boldsymbol{x}) \mathbb{I}_{\{\hat{\pi}_{y'}(\boldsymbol{x}) > \hat{\pi}_y(\boldsymbol{x})\}} + \pi_{(y)}(\boldsymbol{x})}{1 - |\hat{\pi}(\boldsymbol{x}) - \pi(\boldsymbol{x})|} \\
&\leq \frac{2\sqrt{K}\|\hat{\pi}(\boldsymbol{x}) - \pi(\boldsymbol{x})\|_2 + \sum_{y'=1}^{K} \pi_{y'}(\boldsymbol{x}) \mathbb{I}_{\{\hat{\pi}_{y'}(\boldsymbol{x}) > \hat{\pi}_y(\boldsymbol{x})\}} + \pi_{(y)}(\boldsymbol{x})}{1 - \sqrt{K}\|\hat{\pi}(\boldsymbol{x}) - \pi(\boldsymbol{x})\|_2} \\
&\to \sum_{y'=1}^{K} \pi_{y'}(\boldsymbol{x}) \mathbb{I}_{\{\pi_{y'}(\boldsymbol{x}) > \pi_y(\boldsymbol{x})\}} + \pi_{(y)}(\boldsymbol{x}),
\end{aligned}
$$

and

$$
\begin{aligned}
s(\boldsymbol{x}, y; \hat{\pi}) &= \sum_{y'=1}^{K} \hat{\pi}_{(i)}^{\lambda}(\boldsymbol{x}) \mathbb{I}_{\{\hat{\pi}_{(y')}(\boldsymbol{x}) \geq \hat{\pi}_y(\boldsymbol{x})\}} \\
&\geq \frac{-2\|\hat{\pi}(\boldsymbol{x}) - \pi(\boldsymbol{x})\|_2 + \sum_{y'=1}^{K} \pi_{y'}(\boldsymbol{x}) \mathbb{I}_{\{\hat{\pi}_{y'}(\boldsymbol{x}) > \hat{\pi}_y(\boldsymbol{x})\}} + \pi_{(y)}(\boldsymbol{x})}{1 + \|\hat{\pi}_{S_{\boldsymbol{x}}^c}(\boldsymbol{x}) - \pi_{S_{\boldsymbol{x}}^c(\boldsymbol{x})}\|_1} \\
&= \frac{\sum_{y'=1}^{K} \hat{\pi}_{y'}(\boldsymbol{x}) \mathbb{I}_{\{\hat{\pi}_{y'}(\boldsymbol{x}) > \hat{\pi}_y(\boldsymbol{x})\}} + \hat{\pi}_{(y)}(\boldsymbol{x})}{1 - \sum_{i \in S_{\boldsymbol{x}}^c} \hat{\pi}_i(\boldsymbol{x})} \\
&\geq \frac{-2\|\hat{\pi}(\boldsymbol{x}) - \pi(\boldsymbol{x})\|_2^2 + \sum_{y'=1}^{K} \pi_{y'}(\boldsymbol{x}) \mathbb{I}_{\{\hat{\pi}_{y'}(\boldsymbol{x}) > \hat{\pi}_y(\boldsymbol{x})\}} + \pi_{(y)}(\boldsymbol{x})}{1 + \sqrt{K}\|\hat{\pi}(\boldsymbol{x}) - \pi(\boldsymbol{x})\|_2} \\
&\to \sum_{y'=1}^{K} \pi_{y'}(\boldsymbol{x}) \mathbb{I}_{\{\pi_{y'}(\boldsymbol{x}) > \pi_y(\boldsymbol{x})\}} + \pi_{(y)}(\boldsymbol{x}).
\end{aligned}
$$

Combining the inequalities above, the following result holds:

$$s(\boldsymbol{x}, y; \hat{\pi}^{\lambda}) \to s(\boldsymbol{x}, y; \pi) \quad \text{as } n \to \infty.$$

Furthermore, since $\hat{q}^{\lambda}$ and $\hat{q}$ are obtained from the same procedure by applying scores $s(\boldsymbol{x}, y; \hat{\pi}^{\lambda})$ and $s(\boldsymbol{x}, y; \pi)$, respectively, it follows that $\hat{q}^{\lambda} \to \hat{q}$ as $n \to \infty$. Consequently, it concludes that

$$\mathcal{C}_{\text{TCAPS}}(\boldsymbol{x}; \hat{\pi}^{\lambda}, \hat{q}^{\lambda}) \to \mathcal{C}^{\text{oracle}}(\boldsymbol{x}; \pi, q) \quad \text{as } n \to \infty.$$

$\square$

### B.3 PROOF OF THEOREM 8

In this section, we present the proof of marginal coverage for the proposed truncated CP. We begin with the following lemma, which establishes the coverage guarantee for classification tasks in CP.

**Lemma 14** (Coverage Gurantee Angelopoulos & Bates (2021)). *Suppose $\{(\boldsymbol{x}_i, y_i)\}_{i=1}^n$ and $(\boldsymbol{x}_{n+1}, y_{n+1})$ are i.i.d. and let $\mathcal{C}(\boldsymbol{x}; \hat{\pi}, \tau)$ be a set-valued function with estimated probability $\hat{\pi}$ satisfying the nesting property: for any $\tau_1 \leq \tau_2$, we have*

$$\mathcal{C}(\boldsymbol{x}; \hat{\pi}, \tau_1) \subseteq \mathcal{C}(\boldsymbol{x}; \hat{\pi}, \tau_2). \tag{15}$$

*Suppose further that the sets $\mathcal{C}(\boldsymbol{x}; \hat{\pi}, \tau)$ grow to include all labels for large enough $\tau$: for all $\boldsymbol{x} \in \mathbb{R}^d, \mathcal{C}(\boldsymbol{x}; \hat{\pi}, \tau) = \mathcal{Y}$ for some $\tau$. Then for $\hat{\tau}$ defined as the $\lceil (1 - \alpha)(1 + n) \rceil$-largest value of the calibration scores $\{s(\boldsymbol{x}_i, y_i; \hat{\pi})\}_{i=1}^n$, we have the following coverage guarantee:*

$$\mathbb{P}(y_{n+1} \in \mathcal{C}(\boldsymbol{x}_{n+1}; \hat{\pi}, \hat{\tau})) \geq 1 - \alpha.$$

**Theorem 8** (Coverage Guarantee for Truncated CP). *Suppose $\{(\boldsymbol{x}_i, y_i)\}_{i=1}^n$ and $(\boldsymbol{x}_{n+1}, y_{n+1})$ are i.i.d. and let $\mathcal{C}(\boldsymbol{x}_{n+1}; \hat{\pi}^\lambda, \hat{q}^\lambda)$ be the prediction set of $\boldsymbol{x}_{n+1}$ obtained from Algorithm 1 by given confidence level $\alpha$. The following coverage guarantee holds: $\mathbb{P}\left(y_{n+1} \in \mathcal{C}\left(\boldsymbol{x}_{n+1}; \hat{\pi}^\lambda, \hat{q}^\lambda\right)\right) \geq 1 - \alpha$.*

*Proof.* To prove Theorem 8, it suffices to verify the conditions in Lemma 14.

First, note that $\hat{\pi}^\lambda(\cdot)$ is a valid probability distribution over the simplex. The conformal prediction set constructed using the truncated probability estimate $\hat{\pi}^\lambda$ is defined as

$$\mathcal{C}\left(\boldsymbol{x}; \hat{\pi}^\lambda, \hat{q}^\lambda\right) = \left\{y \in \mathcal{Y} : s(\boldsymbol{x}, y; \hat{\pi}^\lambda) \leq \hat{q}^\lambda\right\}.$$

For any $\tau_1 \leq \tau_2$, it follows directly that

$$\left\{y : s(\boldsymbol{x}, y; \hat{\pi}^\lambda) \leq \tau_1\right\} \subseteq \left\{y : s(\boldsymbol{x}, y; \hat{\pi}^\lambda) \leq \tau_2\right\},$$

which verifies the nesting condition required by Lemma 14.

Moreover, setting $\tau = 1$ yields $\mathcal{C}(\boldsymbol{x}; \hat{\pi}^\lambda, 1) = \mathcal{Y}$, thereby satisfying the second condition in Lemma 14. Therefore, the coverage guarantee holds for the truncated conformal prediction method.

$\square$

### B.4 COMPUTATIONAL DETAILS FOR THE EXAMPLES IN SECTION 4.4

In this section, we provide additional computational details for Example 10 and Example 11, including the prediction sets obtained at various quantile levels for different test instances. Before presenting these results, we first present a lemma on the distribution of discrete order statistics.

**Lemma 15** (Distribution of Discrete Order Statistics Casella & Berger (2002)). *Let $X_1, \ldots, X_n$ be i.i.d. random variables drawn from a discrete distribution supported on the ordered set $\{x_1 < x_2 < \cdots < x_r\}$, where $\mathbb{P}(X = x_j) = p_{(j)}$ and $\sum_{j=1}^r p_{(j)} = 1$. Let $X_{(k)}$ denote the $k$-th order statistic of the sample, i.e., the value occupying the $k$-th position after sorting $\{X_1, \ldots, X_n\}$ in increasing order. Then for any $j \in \{1, \ldots, r\}$, the probability that $x_j$ appears at the $k$-th position in the sorted sample is given by*

$$\mathbb{P}(X_{(k)} = x_j) = \sum_{s=0}^{k-1} \sum_{t=k-s}^{n-s} \binom{n}{s, \, t, \, n-s-t} \cdot a^s \cdot b^t \cdot c^{n-s-t}, \tag{16}$$

*where*

$$a = \sum_{i=1}^{j-1} p_{(i)} = \mathbb{P}(X < x_j), \quad b = p_{(j)} = \mathbb{P}(X = x_j), \quad c = \sum_{i=j+1}^{r} p_{(i)} = \mathbb{P}(X > x_j),$$

*and $\binom{n}{s,t,n-s-t} = \frac{n!}{s! \, t! \, (n-s-t)!}$ denotes the multinomial coefficient.*

#### B.4.1 COMPUTATIONAL DETAILS FOR EXAMPLE 10

**Example 10** (APS is Inefficient Under Sparsity Structure). *Let $\boldsymbol{x} \in \mathbb{R}^2$, $\mathcal{Y} = \{1, 2, 3, 4\}$, and $\{(\boldsymbol{x}_i, y_i)\}_{i=1}^n$ be a calibration set, with the underlying true probability defined as: $\pi_{(1)}(\boldsymbol{x}) = 1$, $\pi_{(2)}(\boldsymbol{x}) = \pi_{(3)}(\boldsymbol{x}) = \pi_{(4)}(\boldsymbol{x}) = 0$. Suppose the estimated probability of the input $\boldsymbol{x}$ provided by the model is given as follows: $\hat{\pi}_{(1)}(\boldsymbol{x}) = 1 - 3\xi \cdot \varepsilon$, $\hat{\pi}_{(2)}(\boldsymbol{x}) = \hat{\pi}_{(3)}(\boldsymbol{x}) = \hat{\pi}_{(4)}(\boldsymbol{x}) = \xi \cdot \varepsilon$, where $\varepsilon \sim \mathrm{Bernoulli}(p)$ and $\xi < 0.25$.*

*Then the average prediction set size produced by truncated CP is 1, which is smaller than $1 + 3p \cdot \mathbb{P}(B < \lceil(1 - \alpha)(1 + n)\rceil)$ without truncation, where $B = \sum_{i=1}^n \epsilon_i \sim \mathrm{Binomial}(n, p)$.*

**Detailed Analysis for the Prediction Set Size:** The APS score is computed as $s(\boldsymbol{x}, y; \hat{\pi}) = \sum_{y'=1}^K \hat{\pi}_{y'}(\boldsymbol{x}) \cdot \mathbb{I}_{\{\hat{\pi}_{y'}(\boldsymbol{x}) \geq \hat{\pi}_y(\boldsymbol{x})\}} = \hat{\pi}_{(1)}(\boldsymbol{x}) = 1 - 3\xi \cdot \varepsilon$. The corresponding quantile $\hat{q}$ used for constructing prediction set based on the calibration scores $\{s(\boldsymbol{x}_i, y_i; \hat{\pi})\}_{i=1}^n$ is given by

$$\hat{q} = \begin{cases} 1 - 3\xi, & \text{if } B \geq \lceil(1 - \alpha)(1 + n)\rceil, \\ 1, & \text{otherwise,} \end{cases}$$

where $B = \sum_{i=1}^{n} \varepsilon_i \sim \text{Binomial}(n, p)$. Therefore, the quantile used by APS is either $1$ or $1 - 3\xi$. Assume $\alpha < \xi$, and let $(i)_{\boldsymbol{x}}$ denote the index of the $i$-th largest estimated class probability for input $\boldsymbol{x}$. To ensure the prediction set is non-empty, we define it as the minimal set of labels whose cumulative estimated probabilities exceed or equal $\hat{q}$. When $\hat{q} = 1$, the resulting prediction set is $\{1, 2, 3, 4\}$ with probability $p$, and $\{(1)_{\boldsymbol{x}}\}$ with probability $1 - p$. When $\hat{q} = 1 - 3\xi$, the prediction set simplifies to $\{(1)_{\boldsymbol{x}}\}$. Therefore, the average size of the APS prediction set is $1 + 3p \cdot \mathbb{P}(B < \lceil (1 - \alpha)(1 + n) \rceil)$.

The oracle score is computed as $s(\boldsymbol{x}, y; \pi) = \sum_{y'=1}^{K} \pi_{y'}(\boldsymbol{x}) \cdot \mathbb{I}_{\{\pi_{y'}(\boldsymbol{x}) \geq \pi_y(\boldsymbol{x})\}} = \pi_{(1)}(\boldsymbol{x}) = 1$. As a result, the score for all calibration samples is constant, and the corresponding quantile value is $1$. Thus, the oracle prediction set is $\mathcal{C}^{\text{oracle}}(\boldsymbol{x}; \pi, q) = \{(1)_{\boldsymbol{x}}\}$, and the average prediction set size under the oracle is exactly $1$.

### B.4.2 COMPUTATIONAL DETAILS FOR EXAMPLE 11

**Example 11** (RAPS is Inefficient Under Group Sparsity Structure). *Let $\boldsymbol{x} \in \mathbb{R}^2$, $\mathcal{Y} = \{1, 2, 3, 4\}$, and $\{(\boldsymbol{x}_i, y_i)\}_{i=1}^{n}$ be a calibration set, with the underlying true probability defined as:*

$$\begin{cases} \pi_{(1)}(\boldsymbol{x}) = 1, \quad \pi_{(2)}(\boldsymbol{x}) = \pi_{(3)}(\boldsymbol{x}) = \pi_{(4)}(\boldsymbol{x}) = 0, & \text{if } \boldsymbol{x}[1] > 0, \\ \pi_{(1)}(\boldsymbol{x}) = \frac{2}{3}, \pi_{(2)}(\boldsymbol{x}) = \frac{1}{3}, \quad \pi_{(3)}(\boldsymbol{x}) = \pi_{(4)}(\boldsymbol{x}) = 0, & \text{if } \boldsymbol{x}[1] < 0, \end{cases}$$

*and $\mathbb{P}(\boldsymbol{x}[1] > 0) = p_1$. Suppose the estimated probability of $\boldsymbol{x}$ provided by the model is:*

$$\begin{cases} \hat{\pi}_{(1)}(\boldsymbol{x}) = 1 - 3\xi \cdot \varepsilon, \quad \hat{\pi}_{(2)}(\boldsymbol{x}) = \hat{\pi}_{(3)}(\boldsymbol{x}) = \hat{\pi}_{(4)}(\boldsymbol{x}) = \xi \cdot \varepsilon, & \text{if } \boldsymbol{x}[1] > 0, \\ \hat{\pi}_{(1)}(\boldsymbol{x}) = \frac{2}{3} - \xi \cdot \varepsilon, \ \hat{\pi}_{(2)}(\boldsymbol{x}) = \frac{1}{3} - \xi \cdot \varepsilon, \ \hat{\pi}_{(3)}(\boldsymbol{x}) = \hat{\pi}_{(4)}(\boldsymbol{x}) = \xi \cdot \varepsilon, & \text{if } \boldsymbol{x}[1] < 0, \end{cases}$$

*where $\varepsilon \sim \text{Bernoulli}(p_2)$ and $\xi < \frac{1}{9}$.*

*Then the average prediction set size given by truncated CP is $(q_1' + q_2') + (q_3' + q_4' + q_5')[2 - p_1]$, which is smaller than $q_1' + q_2'[1 + p_2(1 - p_1)] + q_3'[2 - p_1] + q_4'[2 - p_1 + p_1 p_2] + q_5' \times [2 - p_1 + p_2 + p_1 p_2]$ without truncation when $k_{\text{reg}} = 2$, where $q_1', \ldots, q_5' \in (0, 1)$ are probabilities, with their values and more details given in Appendix B.4*

**Detailed Analysis for the Prediction Set Size:** Consider the RAPS setting where $k_{\text{reg}} = 2$ and the hyperparameter $\gamma$ is optimally chosen such that $\gamma > \xi$, in order to minimize the prediction set size. Under this configuration, the possible RAPS scores for a test input $\boldsymbol{x}$ are $\frac{2}{3} - \xi$, $\frac{2}{3}$, $1 - 3\xi$, $1 - 2\xi$, and $1$. The corresponding oracle scores are $\frac{2}{3}, \frac{2}{3}, 1, 1, 1$, respectively. These scores align uniquely between the RAPS and oracle procedures. For instance, a RAPS score of $1 - 3\xi$ corresponds to an oracle score of $1$, and similarly for the remaining cases. There are five possible configurations determined by the quantile $\hat{q}$ selected by RAPS:

- **Case 1:** $\hat{q} = \frac{2}{3} - \xi$. The RAPS prediction set for $\boldsymbol{x}$ is $\{(1)_{\boldsymbol{x}}\}$. The corresponding oracle quantile is $\frac{2}{3}$, yielding the same prediction set $\{(1)_{\boldsymbol{x}}\}$.

- **Case 2:** $\hat{q} = \frac{2}{3}$.
  - If $\boldsymbol{x}[1] > 0$, the RAPS prediction set is $\{(1)_{\boldsymbol{x}}\}$, which coincides with the oracle set.
  - If $\boldsymbol{x}[1] < 0$, the RAPS prediction set is $\{(1)_{\boldsymbol{x}}, (2)_{\boldsymbol{x}}\}$ with probability $p_2$, and $\{(1)_{\boldsymbol{x}}\}$ with probability $1 - p_2$. In both cases, the oracle set remains $\{(1)_{\boldsymbol{x}}\}$.

- **Case 3:** $\hat{q} = 1 - 3\xi$.
  - If $\boldsymbol{x}[1] > 0$, both the RAPS and oracle sets are $\{(1)_{\boldsymbol{x}}\}$.
  - If $\boldsymbol{x}[1] < 0$, the RAPS set is $\{(1)_{\boldsymbol{x}}, (2)_{\boldsymbol{x}}\}$, which matches the oracle set under quantile $q = 1$.

- **Case 4:** $\hat{q} = 1 - 2\xi$.
  - If $\boldsymbol{x}[1] > 0$, the RAPS prediction set is $\{(1)_{\boldsymbol{x}}, (2)_{\boldsymbol{x}}\}$ with probability $p_2$, and $\{(1)_{\boldsymbol{x}}\}$ with probability $1 - p_2$, whereas the oracle set is always $\{(1)_{\boldsymbol{x}}\}$.
  - If $\boldsymbol{x}[1] < 0$, both the RAPS and oracle sets are $\{(1)_{\boldsymbol{x}}, (2)_{\boldsymbol{x}}\}$.

- **Case 5:** $\hat{q} = 1$.

– If $\boldsymbol{x}[1] > 0$, the RAPS prediction set is $\{(1)_{\boldsymbol{x}}, (2)_{\boldsymbol{x}}, (3)_{\boldsymbol{x}}\}$ with probability $p_2$, and $\{(1)_{\boldsymbol{x}}\}$ with probability $1 - p_2$, while the oracle set remains $\{(1)_{\boldsymbol{x}}\}$.

– If $\boldsymbol{x}[1] < 0$, the RAPS prediction set is $\{(1)_{\boldsymbol{x}}, (2)_{\boldsymbol{x}}, (3)_{\boldsymbol{x}}\}$ with probability $p_2$, or $\{(1)_{\boldsymbol{x}}, (2)_{\boldsymbol{x}}\}$ with probability $1 - p_2$, matching the oracle set $\{(1)_{\boldsymbol{x}}, (2)_{\boldsymbol{x}}\}$.

We proceed to compute the corresponding average prediction set size. Based on the above case-wise analysis, it suffices to determine the distribution of the quantile $\hat{q}$. By applying Lemma 15 with $k = \lceil (1 - \alpha)(1 + n) \rceil$, it suffices to evaluate the parameters $a$, $b$ and $c$ in equation 16. The resulting distribution over quantile levels is summarized in Table 4. Consequently, the average prediction set size under RAPS is

$$q_1' + q_2'[1 + p_2(1 - p_1)] + q_3'[2 - p_1] + q_4'[2 - p_1 + p_1 p_2] + q_5'[2 - p_1 + p_2 + p_1 p_2],$$

which strictly exceeds the corresponding size under APS with oracle access to $\pi(\cdot)$, given by

$$(q_1' + q_2') + (q_3' + q_4' + q_5')[2 - p_1].$$

We further consider the case when $k_{\text{reg}} = 1$. Note that the hyperparameter $\gamma$ is optimally selected such that $\gamma > 2\xi$, minimizing the prediction set size. Under this setting, the possible RAPS scores assigned to input $\boldsymbol{x}$ are

$$\frac{2}{3} - \xi < \frac{2}{3} < 1 - 3\xi < 1 < 1 - 2\xi + \lambda < 1 + \lambda,$$

while the corresponding oracle scores are

$$\frac{2}{3}, \quad \frac{2}{3}, \quad 1, \quad 1, \quad 1, \quad \text{and } 1.$$

Each RAPS score is uniquely aligned with an oracle score. Based on these values, there are six distinct cases depending on the quantile $\hat{q}$ selected by RAPS:

- **Case 1:** $\hat{q} = \frac{2}{3} - \xi$. The RAPS prediction set is $\{(1)_{\boldsymbol{x}}\}$, which matches the oracle set obtained at quantile $q = \frac{2}{3}$.

- **Case 2:** $\hat{q} = \frac{2}{3}$.
  - If $\boldsymbol{x}[1] > 0$, both RAPS and oracle sets equal $\{(1)_{\boldsymbol{x}}\}$.
  - If $\boldsymbol{x}[1] < 0$, the RAPS set is $\{(1)_{\boldsymbol{x}}, (2)_{\boldsymbol{x}}\}$ with probability $p_2$, and $\{(1)_{\boldsymbol{x}}\}$ with probability $1 - p_2$; the oracle set remains $\{(1)_{\boldsymbol{x}}\}$.

- **Case 3:** $\hat{q} = 1 - 3\xi$.
  - If $\boldsymbol{x}[1] > 0$, both methods return $\{(1)_{\boldsymbol{x}}\}$.
  - If $\boldsymbol{x}[1] < 0$, the RAPS set is $\{(1)_{\boldsymbol{x}}, (2)_{\boldsymbol{x}}\}$, aligning with the oracle set at quantile 1.

- **Case 4:** $\hat{q} = 1$.
  - If $\boldsymbol{x}[1] > 0$, the RAPS set is $\{(1)_{\boldsymbol{x}}, (2)_{\boldsymbol{x}}\}$ with probability $p_2$, and $\{(1)_{\boldsymbol{x}}\}$ with probability $1 - p_2$; the oracle set remains $\{(1)_{\boldsymbol{x}}\}$.
  - If $\boldsymbol{x}[1] < 0$, both methods yield $\{(1)_{\boldsymbol{x}}, (2)_{\boldsymbol{x}}\}$.

- **Case 5:** $\hat{q} = 1 - 2\xi + \lambda$.
  - If $\boldsymbol{x}[1] > 0$, the RAPS set is deterministically $\{(1)_{\boldsymbol{x}}, (2)_{\boldsymbol{x}}\}$, while the oracle set remains $\{(1)_{\boldsymbol{x}}\}$.
  - If $\boldsymbol{x}[1] < 0$, both methods produce $\{(1)_{\boldsymbol{x}}, (2)_{\boldsymbol{x}}\}$.

- **Case 6:** $\hat{q} = 1 + \lambda$.
  - If $\boldsymbol{x}[1] > 0$, the RAPS set is $\{(1)_{\boldsymbol{x}}, (2)_{\boldsymbol{x}}, (3)_{\boldsymbol{x}}\}$ with probability $p_2$, or $\{(1)_{\boldsymbol{x}}, (2)_{\boldsymbol{x}}\}$ with probability $1 - p_2$; the oracle set remains $\{(1)_{\boldsymbol{x}}\}$.
  - If $\boldsymbol{x}[1] < 0$, the RAPS set is the same as above, while the oracle set is consistently $\{(1)_{\boldsymbol{x}}, (2)_{\boldsymbol{x}}\}$.

Similarly, the computation of the average prediction set size under RAPS reduces to evaluating the parameters $a$, $b$, and $c$ in equation 16. The resulting distribution over quantile levels is summarized in Table 3. Consequently, the average prediction set size under RAPS is

$$q_1 + q_2[1 + p_2(1 - p_1)] + q_3[2 - p_1] + q_4[2 - p_1 + p_1 p_2] + q_5 \cdot 2 + q_6[2 + p_2],$$

which strictly exceeds the corresponding size under APS with oracle access to $\pi(\cdot)$, given by

$$(q_1 + q_2) + (q_3 + q_4 + q_5 + q_6)[2 - p_1].$$

Table 3: Computation Details for $k_{\mathrm{reg}} = 1$

| | $a$ | $b$ | $c$ |
|---|---|---|---|
| $q_1 = \mathbb{P}(\hat{q} = \frac{2}{3} - \xi)$ | $0$ | $\frac{2}{3}(1-p_1)p_2$ | $\frac{1}{3}(1-p_1)(3-2p_2)+p_1$ |
| $q_2 = \mathbb{P}(\hat{q} = \frac{2}{3})$ | $\frac{2}{3}(1-p_1)p_2$ | $\frac{2}{3}(1-p_1)(1-p_2)$ | $\frac{1}{3}(1-p_1)+p_1$ |
| $q_3 = \mathbb{P}(\hat{q} = 1 - 3\xi)$ | $\frac{2}{3}(1-p_1)$ | $p_2 p_1$ | $\frac{1}{3}(1-p_1)+p_1(1-p_2)$ |
| $q_4 = \mathbb{P}(\hat{q} = 1)$ | $\frac{2}{3}(1-p_1)+p_2 p_1$ | $p_1(1-p_2)$ | $\frac{1}{3}(1-p_1)$ |
| $q_5 = \mathbb{P}(\hat{q} = 1 - 2\xi + \gamma)$ | $\frac{2}{3}(1-p_1)+p_1$ | $\frac{1}{3}p_2(1-p_1)$ | $\frac{1}{3}(1-p_2)(1-p_1)$ |
| $q_6 = \mathbb{P}(\hat{q} = 1 + \gamma)$ | $(\frac{2}{3}+\frac{1}{3}p_2)(1-p_1)+p_1$ | $\frac{1}{3}(1-p_1)(1-p_2)$ | $0$ |

Table 4: Computation Details for $k_{\mathrm{reg}} = 2$

| | $a$ | $b$ | $c$ |
|---|---|---|---|
| $q_1' = \mathbb{P}(\hat{q} = \frac{2}{3} - \xi)$ | $0$ | $\frac{2}{3}(1-p_1)p_2$ | $1 - \frac{2}{3}(1-p_1)p_2$ |
| $q_2' = \mathbb{P}(\hat{q} = \frac{2}{3})$ | $\frac{2}{3}(1-p_1)p_2$ | $\frac{2}{3}(1-p_1)(1-p_2)$ | $\frac{1}{3}(1+2p_1)$ |
| $q_3' = \mathbb{P}(\hat{q} = 1 - 3\xi)$ | $\frac{2}{3}(1-p_1)$ | $p_2 p_1$ | $\frac{1}{3}(1+2p_1)-p_2 p_1$ |
| $q_4' = \mathbb{P}(\hat{q} = 1 - 2\xi)$ | $\frac{2}{3}(1-p_1)+p_2 p_1$ | $\frac{1}{3}p_2(1-p_1)$ | $\frac{1}{3}(1-p_2)(1+2p_1)$ |
| $q_3' = \mathbb{P}(\hat{q} = 1)$ | $\frac{1}{3}(1-p_1)(2+p_2)+p_2 p_1$ | $\frac{1}{3}(1-p_2)(1+2p_1)$ | $0$ |

## C PRACTICE DETAILS ON TRUNCATION FRAMEWORK

In this section, we present two practical methods for selecting the truncation level. The first relies on a holdout validation split within the calibration set to obtain the optimal truncation level (Appendix C.1), while the second is a sub-optimal yet more data-efficient approach (Appendix C.2). We then examine in detail the relationship between the first-stage model quality and the optimal truncation level in Appendix C.3. Finally, Appendix C.4 visualizes the score distributions and prediction set sizes before and after truncation, thereby illustrating its effect more clearly.

### C.1 DETAILED INTERPRETATION OF OPTIMAL TRUNCATION LEVELS SELECTION

In this section, we detail the procedure for selecting the optimal truncation level introduced in corollary 5. Specifically, we determine the optimal $\lambda$ via a grid search, using 30% of the calibration set for cross-validation. For each candidate $\lambda$, we compute the average prediction set size and select the largest value that achieves the minimum size, thereby balancing efficiency and coverage. The resulting truncation levels for both simulation (Section 5.1) and real data (Section 5.2) across different models are presented in Table 5. The corresponding "elbow" point, which indicates the optimal truncation level, is illustrated in fig. 6 and Table 6.

Table 5: Truncation levels selection for simulation (left table) and real data experiments (right table).

| GROUP SIZE | GROUP NUM | TRUNCATION |
|---|---|---|
| | 1-1-1 | 0.0130 |
| [10,20,30] | 2-2-2 | 0.0180 |
| | 3-3-3 | 0.0175 |
| | 1-1-1 | 0.0250 |
| [10,30,50] | 2-2-2 | 0.0250 |
| | 3-3-3 | 0.0350 |
| | 1-1-1 | 0.0250 |
| [10,40,70] | 2-2-2 | 0.0300 |
| | 3-3-3 | 0.0250 |

| MODEL | TRUNCATION |
|---|---|
| RESNEXT101 | 0.03 |
| RESNET152 | 0.04 |
| RESNET101 | 0.03 |
| RESNET50 | 0.03 |
| RESNET18 | 0.01 |
| DENSENET161 | 0.04 |
| VGG16 | 0.02 |
| INCEPTION | 0.005 |
| SHUFFLENET | 0.01 |

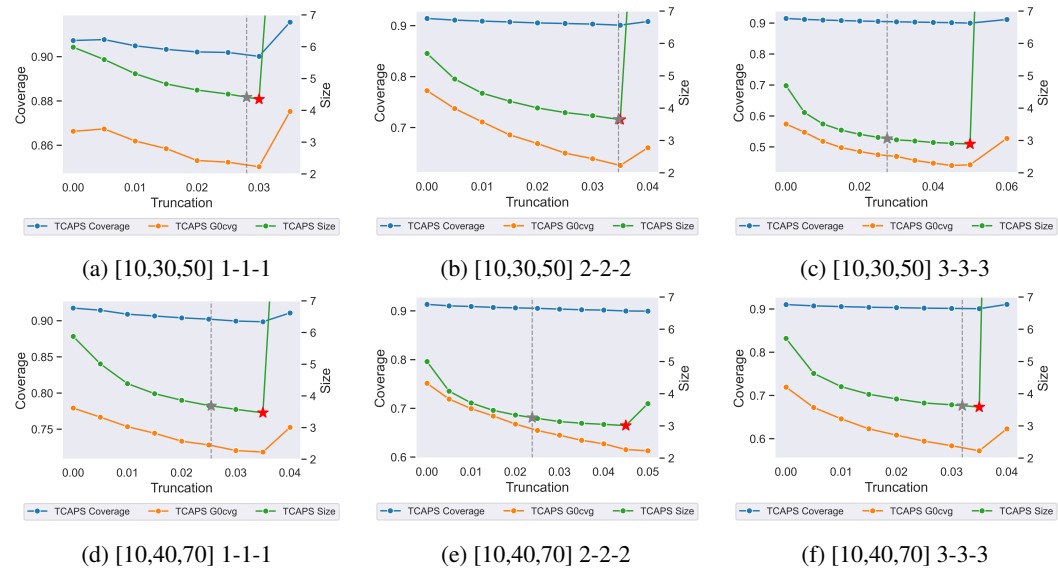

Figure 6: Line plots of group coverage and set size versus sub-optimal truncation level, with overall coverage fixed at $1-\alpha$, across sparsity ratios defined by Group Size and Group Num. The dashed line indicates a sub-optimal truncation level. Gray stars mark the set size at this sub-optimal truncation, while red stars highlight the set size at the optimal truncation. Across varying sparsity ratios, both set size and group coverage exhibit an elbow region.

Table 6: Real data elbow on pretrained ResNet50

| TRUNCATION | 0.00 | 0.02 | 0.04 | 0.06 | 0.08 |
|---|---|---|---|---|---|
| COVERAGE | $93.36 \pm 0.01$ | $90.89 \pm 0.01$ | $90.37 \pm 0.01$ | $93.84 \pm 0.05$ | $100.00 \pm 0.00$ |
| SIZE | $12.29 \pm 0.72$ | $2.58 \pm 0.07$ | $20.76 \pm 36.75$ | $443.48 \pm 454.40$ | $1000.00 \pm 0.00$ |
| GROUP COVERAGE | $80.37 \pm 0.02$ | $80.00 \pm 0.03$ | $78.30 \pm 0.02$ | $87.29 \pm 0.10$ | $100.00 \pm 0.00$ |

## C.2 DETAILED INTERPRETATION OF SUB-OPTIMAL TRUNCATION LEVELS SELECTION

In this section, we provide details on the sub-optimal yet data-efficient truncation level selection method described in remark 6. To avoid allocating additional validation data from the calibration set, we instead exploit the information in the calibration set. The key idea is that the optimal truncation level lies between the signal and the noise, so we aim to estimate this gap directly.

For each calibration sample $(\boldsymbol{x}_i, y_i)$ with $i = 1, \ldots, n$, the true class probability $\pi_{y_i}(\boldsymbol{x}_i)$ represents the largest signal. However, because the first-stage classifier $\hat{\pi}(\cdot)$ is not perfectly accurate, the estimated probability $\hat{\pi}_{y_i}(\boldsymbol{x}_i)$ may sometimes fall within the noise. Consequently, the empirical distribution of $\hat{\pi}_{y_i}(\boldsymbol{x}_i)_{i=1}^n$ carries information about the separation between signal and noise. Since conformal prediction allows for an $\alpha$-level miscoverage, the truncation level should be chosen from an upper quantile of this distribution, *i.e.*, within the range $(1 - \alpha, 1)$.

To formalize this, we introduce the parameter $\psi$ as in Remark 6, with a default choice of $\psi = \alpha/2$. The truncation level is then set to the $(1 - \alpha + \psi)$-quantile of set $\{\hat{\pi}_{y_i}(\boldsymbol{x}_i)\}_{i=1}^n$, which serves as an estimate of the signal–noise boundary. The choice of $\psi$ reflects the quality of the first-stage model: for higher-quality models, a larger $\psi$ is preferred since a greater proportion of $\{\hat{\pi}_{y_i}(\boldsymbol{x}_i)\}_{i=1}^n$ corresponds to signals, whereas for less accurate models, a smaller $\psi$ is advisable due to the higher prevalence of noise. As shown in Figure 6, this sub-optimal selection remains close to the optimal truncation level in practice.

Table 7: Sparse ratio of figure 3

| EPOCH | TRUNCATION | SPARSE RATIO |
|---|---|---|
| 2 | 0.02 | $0.8801 \pm 0.004$ |
| 4 | 0.03 | $0.9235 \pm 0.003$ |
| 8 | 0.05 | $0.9479 \pm 0.002$ |
| 12 | 0.06 | $0.9529 \pm 0.002$ |
| 16 | 0.07 | $0.9580 \pm 0.001$ |
| 30 | 0.08 | $0.9617 \pm 0.001$ |

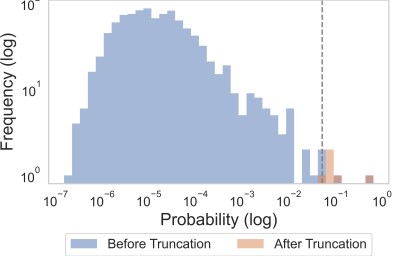 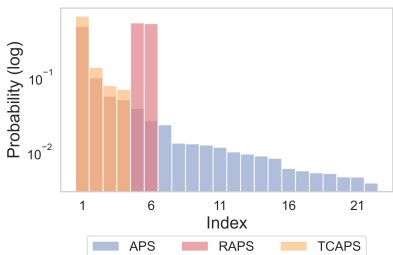

Figure 7: The left figure illustrates the distribution of estimated probabilities before and after truncation, with the dashed line indicating the truncation level, **highlighting a gap between high-probability signals (low proportion) and low-probability noise (high proportion).** The right figure presents the estimated probability scores of the prediction sets obtained by three CP methods. APS and RAPS yield larger set sizes compared to the truncation method (TCAPS), as they do not incorporate sparsity prior information into their algorithms

### C.3 OPTIMAL TRUNCATION LEVEL CORRELATES POSITIVELY WITH MODEL QUALITY

Model quality is critical in CP methods, but the efficiency gains of our truncation framework are relatively insensitive to it. Specifically, high-quality models produce more concentrated probability distributions, whereas poorer models produce noise that forces CP to include extra labels to maintain coverage. Without truncation, such noise prevents the recovery of sparsity. Our truncation approach mitigates this issue by suppressing noise and restoring sparsity. In extreme cases, when the classifier is unreliable and the true label probability is very small, the theoretical guarantees in Appendix B.3 ensure that coverage is preserved, and TCAPS naturally reduces to standard APS as $\lambda \to 0$.

Regarding sparsity, the underlying data distribution determines the true sparsity, while model quality affects how well this sparsity is reflected in predicted probability vector. To quantify this, we calculated the sparsity ratio, *i.e.*, the proportion of zeroed-out labels after truncation for different training epochs. Results are shown in Table 7.

To further quantify the relationship between model quality (training progress) and truncation behavior, we fit scaling curves describing how the optimal truncation threshold and the resulting set size evolve with epochs (see fig. 3). The fitted parameters are as follows: For the truncation value curve: $a_1 = 10.6113$, $b_1 = 1.1719$, and $c_1 = 2.0745$; For the sequence size curve: $a_2 = 0.0796$, $b_2 = 0.1039$, and $c_2 = 0.0839$.

### C.4 DETAILED INTERPRETATION OF THE EFFECT OF TRUNCATION

In this section, we present the distribution of the estimated probability before and after truncation. In the left panel of fig. 7, we show a histogram of the model's softmax-normalized label probabilities. Blue bars correspond to the pre-truncation case (APS), and orange bars to the post-truncation case (TCAPS). The horizontal axis displays estimated probabilities in ascending order, and the vertical axis indicates the frequency within each probability bin. Prior to truncation, a large number of labels yield extremely low probabilities. Moreover, a clear gap emerges between the low-probability "noise" region and the high-probability "signal" region, corresponding to the signal–noise gap described earlier.

In the right panel of fig. 7, we plot the estimated probabilities for each label in the final prediction set output by APS, RAPS, and TCAPS. For APS, these probabilities are simply the model's logits passed through softmax. In RAPS, we see a rise in probabilities starting at index 4, which reflects the penalty added by RAPS at those positions. TCAPS, by contrast, assigns higher probabilities to the first four labels than both APS and RAPS: it truncates the low-probability "noise" in the softmax tail and then renormalizes the remaining values. This comparison shows that adding penalties alone cannot fully eliminate the inefficiency caused by noise, whereas truncation directly reduces the prediction-set size.

From the comparison before and after truncation in Figure 7, it is evident that the improvement from truncation is related to the presence of a sparsity structure. If such a structure is absent in the data, *e.g.*, when the number of classes is not too large, truncation may yield little or no efficiency gain.

## D  Supplementary Experimental Details

In this section, we provide detailed explanations of the previously shown figures and supplement our experiments as follows: Appendix D.1 describes the datasets, model architectures, and experimental environment used throughout our study. Appendix D.2 presents additional results for TCAPS on simulation datasets. Appendix D.3 supplements the omitted results from Table 2 and extends the method to medical imaging data. Appendix D.4 evaluates the TCAPS on not well trained models. Appendix D.5 apply our truncation technique to RAPS, and present the empirical results.

The results in this section demonstrate that the improvement consistently holds across different model qualities and score functions whenever a sparsity structure is present, highlighting the broad applicability of our method. Therefore, for applications targeting efficiency limits—such as deployment on resource-constrained devices—any stable, cost-free performance gain carries substantial practical significance.

### D.1  Experimental Details

**Dataset.** For simulation experiments, we generate data as described in Section 5.1. For real-data experiments, we compare our method with APS and RAPS on the ImageNet-Val dataset Deng et al. (2009). We follow the RAPS protocol and use the ImageNet-Val dataset (50000 images), reserving 10000 images for calibration and the remainder for validation.

**Model Structure.** For the simulation study, we use a simple two-layer feedforward network with a ReLU activation between the layers. For the real-data experiments on ImageNet-Val (Sections 5.2, D.5), we adopt the pretrained architectures from RAPS. For both APS and RAPS methods, temperature scaling (Dabah & Tirer, 2024) is applied by default.

All experiments, including model training and validation, were carried out on a server equipped with a single NVIDIA GeForce RTX 4090 GPU.

### D.2  Additional Experiments on Simulation Data

In this section, we present additional results on simulation data, covering different data-generating settings and a broader range of evaluation metrics. As detailed in Section 5.1, the differing rates of change in set size and group coverage across truncation levels reveal an optimal trade-off between these two metrics. Table 8 (coverage and size) and Table 9 (first-stage classifier quality and group coverage) present the full simulation results, demonstrating that TCAPS achieves higher efficiency than RAPS, attaining superior group coverage under the same coverage constraint.

### D.3  Additional Experiments on Real Data

In this section, we provide additional details on the real-data experiments reported in Table 2 and further extend our study to medical imaging data, a domain where uncertainty plays a critical role.

For the real-data experiments, we extended the evaluation beyond coverage and set size by assessing group coverage, with the results reported in Table 10. For group coverage evaluation, we categorized

Table 8: Full results comparing coverage and prediction-set size for APS, RAPS, and TCAPS on the simulation dataset-across varying sparsity ratios defined by group Size and group Num.

| GROUP | | APS | | RAPS | | TCAPS | |
| --- | --- | --- | --- | --- | --- | --- | --- |
| SIZE | NUM | COVERAGE | SIZE | COVERAGE | SIZE | COVERAGE | SIZE |
| | 1-1-1 | $90.71 \pm 0.01$ | $5.43 \pm 0.10$ | $90.42 \pm 0.01$ | $4.76 \pm 0.32$ | $90.54 \pm 0.01$ | $\mathbf{4.70} \pm 0.05$ |
| [10,20,30] | 2-2-2 | $91.46 \pm 0.01$ | $6.42 \pm 0.05$ | $90.46 \pm 0.01$ | $4.41 \pm 0.09$ | $90.47 \pm 0.01$ | $\mathbf{4.34} \pm 0.06$ |
| | 3-3-3 | $91.57 \pm 0.01$ | $5.49 \pm 0.05$ | $90.37 \pm 0.03$ | $3.51 \pm 0.07$ | $90.50 \pm 0.01$ | $\mathbf{3.45} \pm 0.04$ |
| | 1-1-1 | $90.73 \pm 0.01$ | $5.99 \pm 0.13$ | $90.31 \pm 0.01$ | $4.71 \pm 0.13$ | $90.77 \pm 0.01$ | $\mathbf{4.51} \pm 0.02$ |
| [10,30,50] | 2-2-2 | $91.41 \pm 0.01$ | $5.69 \pm 0.07$ | $90.29 \pm 0.01$ | $4.03 \pm 0.09$ | $90.42 \pm 0.01$ | $\mathbf{3.86} \pm 0.05$ |
| | 3-3-3 | $91.51 \pm 0.01$ | $4.69 \pm 0.02$ | $90.33 \pm 0.01$ | $3.02 \pm 0.03$ | $90.34 \pm 0.01$ | $\mathbf{2.99} \pm 0.03$ |
| | 1-1-1 | $91.75 \pm 0.01$ | $5.88 \pm 0.08$ | $90.23 \pm 0.01$ | $3.84 \pm 0.06$ | $90.22 \pm 0.01$ | $\mathbf{3.70} \pm 0.05$ |
| [10,40,70] | 2-2-2 | $91.33 \pm 0.01$ | $5.00 \pm 0.06$ | $90.30 \pm 0.01$ | $3.19 \pm 0.03$ | $90.37 \pm 0.01$ | $\mathbf{3.14} \pm 0.03$ |
| | 3-3-3 | $91.02 \pm 0.01$ | $5.72 \pm 0.08$ | $90.31 \pm 0.01$ | $3.85 \pm 0.04$ | $90.20 \pm 0.01$ | $\mathbf{3.71} \pm 0.04$ |

Table 9: Comparing group coverage for APS, RAPS, and TCAPS on the simulation dataset—across varying sparsity ratios defined by group size and group count. **Truncation CP not only improves efficiency but also achieves higher group coverage than RAPS.**

| GROUP SIZE | NUM | TOP1 | TOP5 | APS | RAPS | TCAPS |
| --- | --- | --- | --- | --- | --- | --- |
| | 1-1-1 | $58.41 \pm 0.01$ | $89.80 \pm 0.01$ | $75.22 \pm 0.01$ | $72.82 \pm 0.01$ | $73.05 \pm 0.01$ |
| [10,20,30] | 2-2-2 | $60.67 \pm 0.01$ | $89.21 \pm 0.01$ | $81.43 \pm 0.01$ | $76.34 \pm 0.01$ | $76.34 \pm 0.01$ |
| | 3-3-3 | $64.14 \pm 0.01$ | $91.42 \pm 0.01$ | $52.36 \pm 0.01$ | $39.79 \pm 0.01$ | $40.19 \pm 0.01$ |
| | 1-1-1 | $55.46 \pm 0.01$ | $89.13 \pm 0.01$ | $86.63 \pm 0.01$ | $85.38 \pm 0.01$ | $85.31 \pm 0.01$ |
| [10,30,50] | 2-2-2 | $60.23 \pm 0.01$ | $90.52 \pm 0.01$ | $77.25 \pm 0.01$ | $61.02 \pm 0.02$ | $64.98 \pm 0.01$ |
| | 3-3-3 | $66.17 \pm 0.01$ | $92.79 \pm 0.01$ | $57.40 \pm 0.01$ | $44.61 \pm 0.01$ | $45.61 \pm 0.01$ |
| | 1-1-1 | $63.32 \pm 0.01$ | $90.17 \pm 0.01$ | $77.92 \pm 0.01$ | $73.98 \pm 0.01$ | $72.83 \pm 0.01$ |
| [10,40,70] | 2-2-2 | $64.81 \pm 0.01$ | $92.64 \pm 0.01$ | $75.14 \pm 0.01$ | $63.67 \pm 0.01$ | $64.48 \pm 0.01$ |
| | 3-3-3 | $61.62 \pm 0.01$ | $90.81 \pm 0.01$ | $71.95 \pm 0.01$ | $59.28 \pm 0.01$ | $59.44 \pm 0.01$ |

the 1,000 ImageNet classes into 13 broad categories using WordNet Miller (1994) and calculated the coverage of these categories within each prediction set.

Medical imaging is inherently uncertain and a crucial domain for uncertainty quantification. To better demonstrate the practicality of our method, we first applied our truncation strategy to diverse medical imaging datasets: PathMNIST (colon pathology), OrganAMNIST (abdominal CT), TissueMNIST (kidney cortex microscopy), BloodMNIST (blood cell microscopy), and DermaMNIST (dermoscopy). For all experiments, we adopted a unified architecture of five convolutional layers followed by a three-layer MLP (Table 11).

To further validate scalability on a larger dataset, we conducted experiments on a random subset of the NIH Chest X-ray Dataset (released by the National Institutes of Health and Chris Crawford

Table 10: Comparison of Group Coverage for APS, RAPS, and TCAPS on Imagenet-Val.

| METHOD | TOP-1 | TOP-5 | APS | RAPS | TCAPS |
| --- | --- | --- | --- | --- | --- |
| RESNEXT101 | $79.01 \pm 0.01$ | $94.42 \pm 0.01$ | $79.42 \pm 0.02$ | $82.69 \pm 0.01$ | $81.42 \pm 0.02$ |
| RESNET152 | $77.96 \pm 0.01$ | $93.90 \pm 0.01$ | $84.11 \pm 0.04$ | $82.95 \pm 0.01$ | $81.74 \pm 0.02$ |
| RESNET101 | $77.01 \pm 0.01$ | $93.42 \pm 0.01$ | $82.98 \pm 0.04$ | $80.55 \pm 0.01$ | $79.12 \pm 0.02$ |
| RESNET50 | $75.71 \pm 0.01$ | $92.71 \pm 0.01$ | $81.50 \pm 0.04$ | $80.36 \pm 0.02$ | $78.85 \pm 0.02$ |
| RESNET18 | $69.29 \pm 0.01$ | $88.84 \pm 0.01$ | $86.08 \pm 0.01$ | $79.09 \pm 0.01$ | $82.58 \pm 0.01$ |
| DENSENET161 | $76.79 \pm 0.01$ | $93.40 \pm 0.01$ | $78.85 \pm 0.02$ | $81.04 \pm 0.02$ | $78.31 \pm 0.02$ |
| VGG16 | $71.14 \pm 0.01$ | $90.18 \pm 0.01$ | $87.09 \pm 0.03$ | $79.19 \pm 0.01$ | $80.22 \pm 0.01$ |
| INCEPTION | $69.17 \pm 0.01$ | $88.43 \pm 0.01$ | $79.42 \pm 0.05$ | $80.13 \pm 0.01$ | $80.81 \pm 0.02$ |
| SHUFFLENET | $68.86 \pm 0.01$ | $88.10 \pm 0.01$ | $77.32 \pm 0.03$ | $79.73 \pm 0.01$ | $80.73 \pm 0.02$ |

Table 11: Performance comparison of APS, RAPS, and TCAPS on five medical imaging datasets.

| METHOD | APS | | RAPS | | TCAPS | |
|---|---|---|---|---|---|---|
| DATASET | COVERAGE | LENGTH | COVERAGE | LENGTH | COVERAGE | LENGTH |
| PathMNIST | $92.91 \pm 0.01$ | $2.05 \pm 0.16$ | $91.63 \pm 0.01$ | $1.97 \pm 0.09$ | $92.17 \pm 0.01$ | $1.83 \pm 0.09$ |
| OrganAMNIST | $92.48 \pm 0.01$ | $2.52 \pm 0.04$ | $90.25 \pm 0.01$ | $2.32 \pm 0.13$ | $89.67 \pm 0.01$ | $1.98 \pm 0.04$ |
| TissueMNIST | $90.41 \pm 0.01$ | $3.10 \pm 0.03$ | $90.38 \pm 0.01$ | $3.11 \pm 0.05$ | $90.06 \pm 0.01$ | $2.95 \pm 0.04$ |
| BloodMNIST | $92.93 \pm 0.01$ | $3.22 \pm 0.07$ | $91.56 \pm 0.02$ | $3.22 \pm 0.28$ | $92.28 \pm 0.02$ | $3.19 \pm 0.16$ |
| DermaMNIST | $91.95 \pm 0.01$ | $2.54 \pm 0.09$ | $92.22 \pm 0.01$ | $2.62 \pm 0.04$ | $91.45 \pm 0.02$ | $2.50 \pm 0.18$ |

Table 12: Performance comparison of APS, RAPS, and TCAPS on ChestXray

| METHOD | APS | | RAPS | | TCAPS | |
|---|---|---|---|---|---|---|
| MODEL | COVERAGE | LENGTH | COVERAGE | LENGTH | COVERAGE | LENGTH |
| ResNet101 | $90.93 \pm 0.01$ | $6.71 \pm 0.24$ | $90.93 \pm 0.01$ | $6.71 \pm 0.17$ | $90.84 \pm 0.01$ | $6.65 \pm 0.17$ |
| ResNet50 | $90.26 \pm 0.01$ | $5.92 \pm 0.26$ | $90.54 \pm 0.01$ | $6.03 \pm 0.21$ | $90.04 \pm 0.01$ | $5.81 \pm 0.32$ |
| ResNet18 | $90.15 \pm 0.02$ | $5.64 \pm 0.50$ | $90.15 \pm 0.01$ | $5.67 \pm 0.55$ | $89.94 \pm 0.02$ | $5.58 \pm 0.64$ |

on Kaggle), which contains 5,606 images with corresponding labels across 15 disease categories (Table 12). To simplify the experimental setup, we excluded multi-label samples, retained 4,626 single-label cases, and split them into 50% training, 25% calibration, and 25% testing. We use ResNet18, ResNet50, and ResNet101 as our models, all of which are trained from scratch.

### D.4    ADDITIONAL EXPERIMENTS ON NOT WELL TRAINED MODEL

In this section, we present experiments demonstrating that our truncation framework remains effective even with a moderately trained first-stage classifier. While our framework can benefit from a high-quality base classifier, it can still provide meaningful efficiency gains when applied to moderately trained models, making it broadly applicable in practice. In particular, even when the classifier is not well trained (Top 1 Accuracy is not so high), applying truncation can still help reduce the prediction set length, as demonstrated in Table 13 and Table 14.

### D.5    EXTENDING TRUNCATION TECHNIQUES TO RAPS

In this section, we further extend our framework by applying truncation to RAPS (see Table 15). Since the truncation framework already removes the majority of noise, applying RAPS to the truncated scores yields no significant additional improvement.

Table 13: Coverage and prediction-set size for a not well trained model on simulation data

| GROUP | | APS | | RAPS | | TCAPS | |
|---|---|---|---|---|---|---|---|
| SIZE | NUM | COVERAGE | SIZE | COVERAGE | SIZE | COVERAGE | SIZE |
| | 1–1–1 | $90.39 \pm 0.01$ | $13.51 \pm 0.15$ | $90.51 \pm 0.01$ | $13.65 \pm 0.23$ | $90.50 \pm 0.01$ | $13.44 \pm 0.18$ |
| [10–20–30] | 2–2–2 | $90.46 \pm 0.01$ | $16.22 \pm 0.21$ | $90.19 \pm 0.01$ | $13.80 \pm 0.44$ | $89.91 \pm 0.01$ | $12.60 \pm 0.35$ |
| | 3–3–3 | $90.52 \pm 0.01$ | $11.21 \pm 0.12$ | $89.93 \pm 0.01$ | $8.05 \pm 0.13$ | $89.96 \pm 0.01$ | $7.64 \pm 0.22$ |
| | 1–1–1 | $90.09 \pm 0.01$ | $15.45 \pm 0.31$ | $90.17 \pm 0.01$ | $14.56 \pm 0.33$ | $89.97 \pm 0.01$ | $14.17 \pm 0.21$ |
| [10–30–50] | 2–2–2 | $90.53 \pm 0.01$ | $11.73 \pm 0.07$ | $90.15 \pm 0.01$ | $8.70 \pm 0.01$ | $90.19 \pm 0.01$ | $8.21 \pm 0.11$ |
| | 3–3–3 | $90.48 \pm 0.01$ | $9.88 \pm 0.13$ | $90.10 \pm 0.01$ | $6.75 \pm 0.10$ | $89.94 \pm 0.01$ | $6.19 \pm 0.05$ |
| | 1–1–1 | $90.20 \pm 0.01$ | $12.11 \pm 0.21$ | $90.02 \pm 0.01$ | $10.25 \pm 0.39$ | $90.38 \pm 0.01$ | $12.04 \pm 2.85$ |
| [10–40–70] | 2–2–2 | $90.44 \pm 0.01$ | $9.91 \pm 0.05$ | $90.16 \pm 0.01$ | $6.63 \pm 0.10$ | $90.14 \pm 0.01$ | $8.05 \pm 3.86$ |
| | 3–3–3 | $90.33 \pm 0.01$ | $15.36 \pm 0.14$ | $90.12 \pm 0.01$ | $10.05 \pm 0.09$ | $90.11 \pm 0.01$ | $9.31 \pm 0.08$ |

Table 14: Comparing group coverage for a not well trained model on simulation data

| GROUP SIZE | NUM | TOP1 | TOP5 | APS | RAPS | TCAPS |
|---|---|---|---|---|---|---|
| | 1–1–1 | $41.07 \pm 0.01$ | $72.84 \pm 0.01$ | $71.32 \pm 0.01$ | $71.42 \pm 0.01$ | $71.74 \pm 0.01$ |
| [10–20–30] | 2–2–2 | $46.44 \pm 0.01$ | $76.09 \pm 0.01$ | $79.39 \pm 0.01$ | $74.47 \pm 0.05$ | $73.39 \pm 0.01$ |
| | 3–3–3 | $54.27 \pm 0.01$ | $83.01 \pm 0.01$ | $55.04 \pm 0.02$ | $31.93 \pm 0.02$ | $33.38 \pm 0.01$ |
| | 1–1–1 | $38.63 \pm 0.01$ | $72.26 \pm 0.01$ | $85.50 \pm 0.01$ | $84.44 \pm 0.01$ | $84.77 \pm 0.01$ |
| [10–30–50] | 2–2–2 | $49.46 \pm 0.01$ | $81.03 \pm 0.01$ | $67.02 \pm 0.01$ | $52.99 \pm 0.01$ | $55.27 \pm 0.01$ |
| | 3–3–3 | $55.06 \pm 0.01$ | $85.34 \pm 3.03$ | $47.13 \pm 0.01$ | $34.77 \pm 0.01$ | $36.29 \pm 0.01$ |
| | 1–1–1 | $50.67 \pm 0.01$ | $79.65 \pm 0.01$ | $62.06 \pm 0.01$ | $64.37 \pm 0.03$ | $61.93 \pm 0.02$ |
| [10–40–70] | 2–2–2 | $54.67 \pm 0.01$ | $85.07 \pm 0.01$ | $68.47 \pm 0.01$ | $57.82 \pm 0.01$ | $58.77 \pm 0.02$ |
| | 3–3–3 | $50.46 \pm 0.01$ | $81.55 \pm 0.01$ | $70.30 \pm 0.01$ | $56.97 \pm 0.01$ | $58.53 \pm 0.01$ |

Table 15: Comparison of coverage, set size, and group coverage for TCAPS and TCRAPS, which applying truncation technique within the RAPS framework. Since truncation already effectively removes noise, adding the RAPS penalty does not yield significant efficiency improvement.

| GROUP | | TCAPS | | TCRAPS | | GROUP COVERAGE | |
|---|---|---|---|---|---|---|---|
| SIZE | NUM | COVERAGE | SIZE | COVERAGE | SIZE | TCAPS | TCRAPS |
| | 1-1-1 | $90.54 \pm 0.01$ | $4.70 \pm 0.05$ | $90.53 \pm 0.01$ | $\mathbf{4.69} \pm 0.28$ | $73.05 \pm 0.01$ | $72.88 \pm 0.01$ |
| [10,20,30] | 2-2-2 | $90.47 \pm 0.01$ | $\mathbf{4.37} \pm 0.06$ | $90.49 \pm 0.01$ | $4.39 \pm 0.06$ | $76.34 \pm 0.01$ | $76.45 \pm 0.01$ |
| | 3-3-3 | $90.34 \pm 0.01$ | $\mathbf{2.99} \pm 0.03$ | $90.33 \pm 0.01$ | $3.33 \pm 0.03$ | $40.19 \pm 0.01$ | $36.10 \pm 0.01$ |
| | 1-1-1 | $90.34 \pm 0.01$ | $\mathbf{3.00} \pm 0.10$ | $90.00 \pm 0.01$ | $4.41 \pm 0.10$ | $85.31 \pm 0.01$ | $85.11 \pm 0.01$ |
| [10,30,50] | 2-2-2 | $90.42 \pm 0.01$ | $\mathbf{3.86} \pm 0.05$ | $90.22 \pm 0.01$ | $4.04 \pm 0.57$ | $64.98 \pm 0.01$ | $62.69 \pm 0.02$ |
| | 3-3-3 | $90.16 \pm 0.01$ | $4.46 \pm 0.12$ | $90.01 \pm 0.01$ | $\mathbf{3.07} \pm 0.47$ | $45.61 \pm 0.01$ | $42.65 \pm 0.01$ |
| | 1-1-1 | $90.22 \pm 0.01$ | $3.70 \pm 0.05$ | $90.15 \pm 0.01$ | $\mathbf{3.62} \pm 0.09$ | $72.83 \pm 0.01$ | $72.81 \pm 0.01$ |
| [10,40,70] | 2-2-2 | $90.37 \pm 0.01$ | $3.14 \pm 0.03$ | $90.05 \pm 0.01$ | $\mathbf{2.98} \pm 0.02$ | $64.48 \pm 0.01$ | $62.01 \pm 0.01$ |
| | 3-3-3 | $90.20 \pm 0.01$ | $\mathbf{3.71} \pm 0.04$ | $90.14 \pm 0.01$ | $3.95 \pm 0.55$ | $59.44 \pm 0.01$ | $57.61 \pm 0.01$ |

# E  THE USE OF LARGE LANGUAGE MODELS

In this work, Large Language Models (LLMs) are employed under our supervision to support language polishing and grammar correction. Suggestions provided by the models are carefully reviewed and selectively adopted, ensuring accuracy, consistency, and academic integrity.

