# OpenReview forum: "Truncated Conformal Prediction:  A Sparsity-Aware Framework for Classification"
_ICLR.cc/2026/Conference — Submitted to ICLR 2026_

### Official Review · Reviewer_6xdA · 2025-10-26

**Soundness:** 3
**Presentation:** 3
**Contribution:** 3
**Rating:** 4
**Confidence:** 4

**Summary:**

The paper improves conformal prediction (CP) efficiency by encoding label sparsity: before CP, it truncates tiny class probabilities and renormalizes the vector, then applies standard CP (theory centers on APS). This yields a U-shaped trade-off between set size and the truncation level, suggesting an elbow rule for choosing the threshold. Under a consistent base model, truncated CP asymptotically matches the oracle set predictor—typically unattainable without truncation. Experiments show smaller prediction sets at the same coverage, with larger gains as the classifier improves.

**Strengths:**

1. The paper proposes a simple, intuitive, and effective post-hoc method to improve prediction efficiency in conformal classification.
2. It provides theoretical guarantees for coverage and efficiency, and offers explicit guidance for selecting the truncation threshold.
3. Extensive experiments show consistent empirical gains and supply practical guidelines, further strengthening the contribution.

**Weaknesses:**

1. The experiments use a limited set of baselines, despite numerous existing approaches that enhance prediction efficiency in conformal classification.
2. The method may be limited to classification settings with a sparsity prior.

**Questions:**

1. The experimental baselines are limited. Please include stronger comparators [1,2] in the experiments.
2. Remark 6 introduces a heuristic for choosing a sub-optimal truncation level from the calibration set. Please provide practical guidance on when to use this heuristic instead of a validation sweep, how to set , and its sensitivity to model quality and sample size.
3. The method is well-motivated for multiclass classification with sparse label support. Will this method fall short in LLM-style generation classification or settings with dense output distributions (large vocabularies, little inherent sparsity)?
4. Do the authors have ideas for extending this approach to continuous responses (regression CP) or to settings with distribution shift to broaden the impact?

[1] Trustworthy Classification through Rank-Based Conformal Prediction Sets.
[2] Conformal Prediction for Deep Classifier via Label Ranking.

---

### Official Review · Reviewer_gNdq · 2025-10-29

**Soundness:** 2
**Presentation:** 3
**Contribution:** 2
**Rating:** 2
**Confidence:** 4

**Summary:**

This paper proposes a sparsity-aware truncation–normalization framework to improve the efficiency of conformal prediction (CP) for classification. The method zeroes out small probabilities (tail mass) and renormalizes the remaining mass, effectively reallocating probability from low-probability labels to higher-probability ones. The authors provide theoretical analyses (U-shaped size–truncation behavior, asymptotic oracle recovery with APS) and demonstrate empirical gains over APS and RAPS.

**Strengths:**

1. Clarity of exposition: Figure 1 is an effective high-level illustration. The method is easy to understand even without delving into formal details.
2. Empirical performance: The proposed approach consistently outperforms APS and RAPS in set size while maintaining nominal marginal coverage across synthetic and real datasets, including large-scale ImageNet models.

**Weaknesses:**

1. Limited conceptual novelty: Although the operator differs from RAPS, the high-level idea is similar—suppress tail mass and reallocate weight to more plausible labels. The contribution feels incremental at a conceptual level.

2. Conditional validity not addressed: Like APS and RAPS, the method does not provide conditional guarantees. Given the recent emphasis on conditional validity, this is a notable gap. Moreover, APS is designed to trade efficiency for better conditional behavior, so beating APS on efficiency alone is not surprising nor necessarily the right baseline for that comparison.

3. Baselines and positioning: APS and RAPS are now relatively dated. More recent work (e.g., [1]) focuses on conditional guarantees, and efficiency-aware post-selection frameworks (e.g., [2]) may subsume both RAPS and the proposed truncation as special cases. Introducing a new hyperparameter $\lambda$ to optimize efficiency aligns with [2], so the gains are somewhat expected unless contrasted against stronger contemporary baselines or integrated within modern post-selection CP pipelines.

[1] Gibbs, Isaac, John J. Cherian, and Emmanuel J. Candès. “Conformal prediction with conditional guarantees.” JRSS-B (2025): qkaf008.

[2] Liang, Ruiting, Wanrong Zhu, and Rina Foygel Barber. “Conformal prediction after efficiency-oriented model selection.” arXiv:2408.07066 (2024).

**Questions:**

Table 2 APS coverage: The APS coverage reported in Table 2 is around 93–94%, whereas the nominal level is 90%. This suggests a potential implementation or calibration issue.

---

### Official Review · Reviewer_nUW7 · 2025-10-29

**Soundness:** 3
**Presentation:** 2
**Contribution:** 3
**Rating:** 4
**Confidence:** 4

**Summary:**

The paper introduces a method to improve the efficiency of conformal prediction by truncating class probabilities below a certain threshold and renormalizing the remaining values to form a valid probability distribution. The authors propose strategies for setting this threshold, both with and without a validation set, by analyzing the U-shaped behavior of prediction set size as a function of the threshold. They provide theoretical insights explaining why this pattern emerges and show that, when properly tuned, the truncation operator can recover optimal prediction sets. Empirical results on both synthetic and real datasets demonstrate improved efficiency compared to standard conformal prediction.

**Strengths:**

The core idea of the paper is simple, effective, and easy to apply to a wide range of conformal prediction tasks in classification. Because of its practicality and broad relevance, it has the potential to make a strong impact both within the conformal prediction community and in downstream applications. The method is well-supported by both theoretical analysis and empirical results.

**Weaknesses:**

The main weakness of the paper lies in its presentation. The notation is often heavy and difficult to follow, which makes some of the theoretical results hard to interpret. For example, I was unable to fully understand Theorem 7 (see questions below). Since the theoretical contributions are a key part of the paper, my score reflects the need for clearer exposition. However, I am open to increasing it if the authors clarify the results and improve the overall presentation.

### Minor issues
- $\psi$ is never properly defined in the main text and the discussion in Remark 6 is a bit confusing because of that.
- At the end of page 19, it says “The results indicate that truncation preserves the relative ranking of class probabilities for most samples”. I believe the intended meaning is that the ranking of the correct class scores across calibration points remains largely unchanged. However, the current phrasing could be misread as referring to the ranking over classes within a single sample, which wouldn’t be affected by truncation. I’d suggest rewording to make it clear that you’re referring to the first case.

**Questions:**

1. In Theorem 4, it says that if $\lambda \geq \frac{1}{K}$ then $L_0 \geq L_\lambda$, which confuses me. In that case, shouldn’t $L_\lambda$ be larger? The text gives $\lambda = 1$ as an example of a poor choice that leads to large prediction sets including all labels.
2. In Remark 6, since the number of classes $K$ plays an important role in Theorem 4, shouldn’t it also be taken into account when choosing $\lambda$ without a validation set?
3. Figure 2. What exactly does the orange line represent? I’d suggest adding more detail to the caption to clarify the curves. I assume it refers to the coverage of a specific group, since the text later mentions: “As shown in Fig. 2, both size and coverage curves exhibit elbow points as the truncation level increases.” Since truncation essentially defines a new score function, I’d expect coverage to remain constant, as seen in the blue line. However, it’s unclear why truncation would affect specific groups in such a monotonic way. Could you explain?
4. Theorem 7. To be honest, I could not understand what this theorem is trying to show in the first place. I would appreciate if the authors could elaborate on that further. The proof was not clear either, especially the third step. The notation there is a bit confusing. Also, why is it reasonable to assume that $\hat \pi(x)$ approaches $\pi(x)$ as the calibration size increases? If the estimator is fixed during calibration, why would it converge to the true probability asymptotically in $n$? Or is $n$ here referring to the training set size rather than the calibration set?
5. Extensions. The paper seems to be written with split conformal prediction in mind, but could it generalize to full conformal prediction too? Would the results also hold for regression under a regression-as-classification framework [1]?

### References

[1] Guha, Etash Kumar, et al. "Conformal Prediction via Regression-as-Classification." The Twelfth International Conference on Learning Representations.

---

### Official Review · Reviewer_tAjo · 2025-10-31

**Soundness:** 2
**Presentation:** 3
**Contribution:** 2
**Rating:** 4
**Confidence:** 3

**Summary:**

The paper proposes Truncated Conformal Prediction (TCP), a post-hoc framework that improves the efficiency of conformal prediction (CP) in classification tasks. The method introduces a truncation-normalization operator that enforces sparsity on the predicted class probabilities by zeroing out small probability values and renormalizing the remaining mass. This sparsity-aware transformation can be applied to any score-based CP method (e.g., APS, RAPS). Theoretically, the authors show that the prediction-set size exhibits a U-shaped dependency on the truncation level $\lambda$, and there exists an optimal $\lambda^*$ yielding asymptotically minimal set size while maintaining marginal coverage. Empirically, experiments on synthetic data, ImageNet-Val, and several medical imaging benchmarks demonstrate consistent reductions in prediction-set sizes.

**Strengths:**

1. The paper addresses an important and persistent issue in conformal prediction: inefficiency of prediction sets. The proposed truncation approach is simple, intuitive, and compatible with existing CP pipelines without retraining models.
2. The method is extensively evaluated on multiple large-scale datasets (ImageNet, MedMNIST) with diverse backbones, consistently improving efficiency at comparable coverage.
3. Figures and algorithms are easy to follow; the “elbow” behavior of $\lambda$ is well visualized and intuitively matches theory.

**Weaknesses:**

1. The central assumption that the true conditional distribution $p(y|x)$ is sparse is not theoretically justified and may not hold in fine-grained or multi-label tasks.
2. The "U-shape” theorem lacks explicit regularity conditions on the score function or on the truncation mapping.
3. The “asymptotic optimality” (Theorem 7) is tautological, where λ* equals the unknown signal–noise boundary δ, so the theorem holds by construction rather than by learnable conditions.
4. Coverage preservation (Theorem 8) assumes marginal exchangeability but ignores potential violations when truncation changes score ordering.
5. The truncation level λ is chosen empirically using a hold-out set; the paper provides no principled or learnable rule for λ selection. This limits reproducibility and weakens the claimed theoretical optimality.
6. The work does not compare against training-time sparsity approaches such as sparsemax or entmax, which also produce sparse probability vectors and could yield similar gains without post-hoc truncation.
7. The coverage guarantee holds only in the marginal sense. Conditional coverage may degrade when sparsity varies across samples; the paper does not examine this.

**Questions:**

1. Could you provide quantitative evidence that softmax outputs indeed destroy sparsity (e.g., histogram of small-probability mass before/after truncation)?
2. Could you include sparsemax / entmax / temperature-scaling baselines to disentangle the benefit of truncation from calibration effects?
3. Could you evaluate conditional coverage, class-conditional coverage or group-conditional coverage to assess fairness and robustness?
4. Could you discuss or learn an adaptive λ(x) to handle heterogeneity in sparsity across samples?

---

### Meta-Review · Area_Chair_hKjv · 2026-01-04

**Summary:**

All the reviews are not positive and they raised many issues related to significance, novelty and evaluation. There is no rebuttal from the authors.

**Reviewer Concerns:**

There is no rebuttal from the authors.

**Reviewer Scores:**

There is no rebuttal from the authors.

---

### Decision · Program_Chairs · 2026-01-26

Reject